# Quassinoid analogs with enhanced efficacy for treatment of hematologic malignancies target the PI3Kγ isoform

Yonggang Pei [1], Nicky Hwang[2], Fengchao Lang[1], Lanlan Zhou[3], Josiah Hiu-yuen Wong[1], Rajnish Kumar Singh[1,4], Hem Chandra Jha [1,5], Wafik S. El-Deiry[3], Yanming Du[2] & Erle S. Robertson [1✉]

Development of novel PI3K inhibitors is an important strategy to overcome their resistance and poor tolerability in clinical trials. The quassinoid family member Brusatol shows specific inhibitory activity against hematologic malignances. However, the mechanism of its anti-cancer activity is unknown. We investigated the anti-cancer activity of Brusatol on multiple hematologic malignancies derived cell lines. The results demonstrated that the PI3Kγ isoform was identified as a direct target of Brusatol, and inhibition was dramatically reduced on cells with lower PI3Kγ levels. Novel synthetic analogs were also developed and tested in vitro and in vivo. They shared comparable or superior potency in their ability to inhibit malignant hematologic cell lines, and in a xenograft transplant mouse model. One unique analog had minimal toxicity to normal human cells and in a mouse model. These new analogs have enhanced potential for development as a new class of PI3K inhibitors for treatment of hematologic malignancies.

---

[1] Departments of Otorhinolaryngology-Head and Neck Surgery, and Microbiology, and the Tumor Virology Program, Abramson Cancer Center, Perelman School of Medicine at the University of Pennsylvania, Philadelphia, PA, USA. [2] Baruch S. Blumberg Institute, Hepatitis B Foundation, Doylestown, PA, USA. [3] Department of Pathology and Laboratory Medicine, Department of Medical Science, Warren Alpert Medical School, Brown University, Providence, RI, USA. [4] Present address: Molecular Biology Unit, Institute of Medical Sciences, Banaras Hindu University, Varanasi, UP, India. [5] Present address: Discipline of Biosciences and Biomedical Engineering, Indian Institute of Technology Indore, Simrol Campus, Simrol, MP, India. ✉email: erle@pennmedicine.upenn.edu

Phosphoinositide 3-kinases (PI3Ks) are involved in a broad range of physiological processes, and their elevated activities are typically identified as a hallmark of many human cancers[1]. Development of small-molecule inhibitors selectively targeting PI3Ks is an extremely promising strategy as the PI3Ks pathway is associated with many processes that drive oncogenesis. Three PI3K inhibitors, Idelalisib, Copanlisib, Duvelisib, are currently FDA approved for treating relapsed or refractory chronic lymphocytic leukemia (CLL), small lymphocytic lymphoma (SLL), or indolent lymphoma[2–4]. Another PI3Kγ inhibitor IPI-549 which overcomes resistance to immune checkpoint blockade (ICB), and reshapes the tumor microenvironment to promote cytotoxic-T-lymphocyte-associated tumor regression is also progressing in clinical trials[5,6]. These inhibitors demonstrate the potential of targeting the PI3K protein family members for treatment of numerous types of cancers[7]. However, a major obstacle limiting their success in clinical trials is the resistance and poor tolerability to PI3K inhibitors[8]. Therefore, proposed solutions for improving their impact as cancer therapeutics include the development of more specific PI3K inhibitors and the amelioration of their associated toxicities[8]. Ongoing clinical trials with PI3K targeted therapies combined with other therapies will provide evidence to support their use as therapeutic agents for treating human cancers[7].

The quassinoid family of over 150 members is known to have anti-cancer activities[9]. Brusatol, isolated from the plant *Brucea Javanica*, has anti-cancer activity but has not been previously tested in human patients[10,11]. Its toxicity has been a major obstacle to clinical applications, and its anti-cancer mechanisms also need further investigation. One study showed that Brusatol can be used as an adjuvant chemotherapeutic drug in A549 lung cancer cells derived xenografts by inhibiting the Nrf2 signaling pathway[12]. However, the results suggested that inhibition was directly through targeting the translation of cap-dependent and cap-independent proteins rather than Nrf2[13]. Yet, another study showed that Brusatol functions as a protein synthesis inhibitor in a manner independent of Nrf2[14]. Nevertheless, the mechanism of its anti-cancer activities is not fully understood, even though some studies have recognized Brusatol as an Nrf2 specific inhibitor. This hinders the development and use of quassinoids family members as cancer therapeutics.

Our current study shows that Brusatol has strong inhibitory activities on multiple types of hematologic malignancies. Further, we have identified its direct targets, and the effects on their downstream signaling activities. More specifically, Brusatol targeted PI3Kγ to inhibit PI3K/AKT signaling pathway. This led to the induction of cell death in many cancer cell lines. Furthermore, synthetic analogs developed, showed comparable or superior bioactivity, and one unique analog had potent inhibitory activities and reduced toxicity in cell culture as well as a preclinical xeno-transplant mice model. This demonstrates that these analogs have strong potential to be further developed for treatment of hematologic malignancies and other cancers with elevated PI3Kγ levels.

## Results

### Brusatol inhibits viability of hematologic malignancy derived cells.
The quassinoid family member Brusatol has been previously demonstrated to show inhibitory activities towards the growth of some leukemias[10,15,16]. To investigate its effects on EBV-associated lymphomas and other types of lymphomas, we monitored the viability of a Burkitt's lymphoma (BL) cell line Raji and another Epstein Barr virus (EBV)-transformed lymphoblastoid cell line (LCL1) when treated with Brusatol[17]. Our results showed that Brusatol effectively inhibited the growth of the BL cell line Raji even at increasing cell concentrations (Fig. 1a). Furthermore,

it also exhibited strong inhibitory effects on the EBV-transformed LCL1 cells (Fig. 1b). The half-maximal inhibitory concentration (IC50) was determined for both Raji and LCL1 cells. The results showed an effective IC50 of between 2 nM and 6 nM for both cell lines (Fig. 1c).

To determine the ability of Brusatol to inhibit other types of lymphomas and leukemias, we tested its inhibition of a number of different human hematologic malignancies derived cell lines (Supplementary Table, Supplementary Fig. 1a). The results demonstrated that Brusatol had broad and potent inhibitory effects on the majority of these cell lines in addition to EBV-associated lymphomas, which include lymphoma, leukemia, and multiple myeloma cell lines (Fig. 1d). Brusatol also inhibited hematologic malignant cells obtained from three independent patient-derived xenografts (PDX) (Fig. 1e). These results provide new evidence as to the clinical potential of Brusatol for treatment of a broad range of hematologic malignancies. Furthermore, cell cycle analyses showed that Brusatol strongly disrupted the cell cycle phases, and induced cell death 72 h post-treatment, which was consistent with the results of cell viability assays in EBV-transformed LCLs (Fig. 1f). Its inhibitory effects were associated with inhibition of cell cycle progression with a 3.5 to 4-fold increase in subG1 population in cells treated with Brusatol (Fig. 1f). These studies demonstrated that Brusatol inhibited the growth of EBV-positive as well as multiple hematologic malignant cell lines in vitro, and has strong potential as a candidate small-molecule therapeutic agent for treatment of hematologic malignancies.

### Brusatol specifically targets the PI3K/AKT signaling pathway.
Brusatol exhibited significant inhibition of the hematologic malignancies derived cell lines (Fig .1). However, the mechanism of its anti-cancer activities has not previously explored. To address this, RNA-Seq analysis was performed with Brusatol-treated EBV-transformed LCLs to characterize the transcription reprogramming events. Analysis of the results showed that SLC22A1, a twelve-membrane cation transporter[18], was highly upregulated and may be linked to the elimination of Brusatol. CTNNA1, the catenin family member involved in the connection of cadherin to actin filament[19], was dramatically downregulated. A list of twelve genes shown was the most significantly altered from our transcription analyses (Fig. 2a). Other important genes involved in a broad range of cellular pathways were also affected (Supplementary Fig. 1b, c). The regulated genes and their associated pathways are shown (Fig. 2a, b). Specifically, the results showed that EIF2 signaling, protein ubiquitination, and mTOR signaling, as well as regulation of eIF4 and p70S6K signaling, were critical pathways involved in Brusatol-mediated inhibition (Fig. 2b, and Supplementary Fig. 1b). These pathways are tightly associated with the regulation of protein synthesis[20–22]. Therefore, it is reasonable that Brusatol was recognized as an inhibitor of protein synthesis[14,23]. Additionally, the NRF2-mediated oxidative stress response was also linked to Brusatol treatment (Fig. 2b). This supports a previous study which showed that Brusatol inhibited the NRF2 pathway[12]. NF-κB and c-Myc-associated pathways were also previously linked to Brusatol-associated inhibition[24,25]. However, no definitive results have supported their role as direct effectors of Brusatol inhibition although it is certainly possible that they may have an indirect role[25,26].

To explore the results of our RNA-Seq, upstream regulator analysis using the Ingenuity Pathway Analysis (IPA) program was performed[27]. Analysis of the mRNA profile of Brusatol-treated cells identified critical upstream regulators, which include specific PI3K family members, *JUP*, *TP73*, and *TP53* (Fig. 2c, and

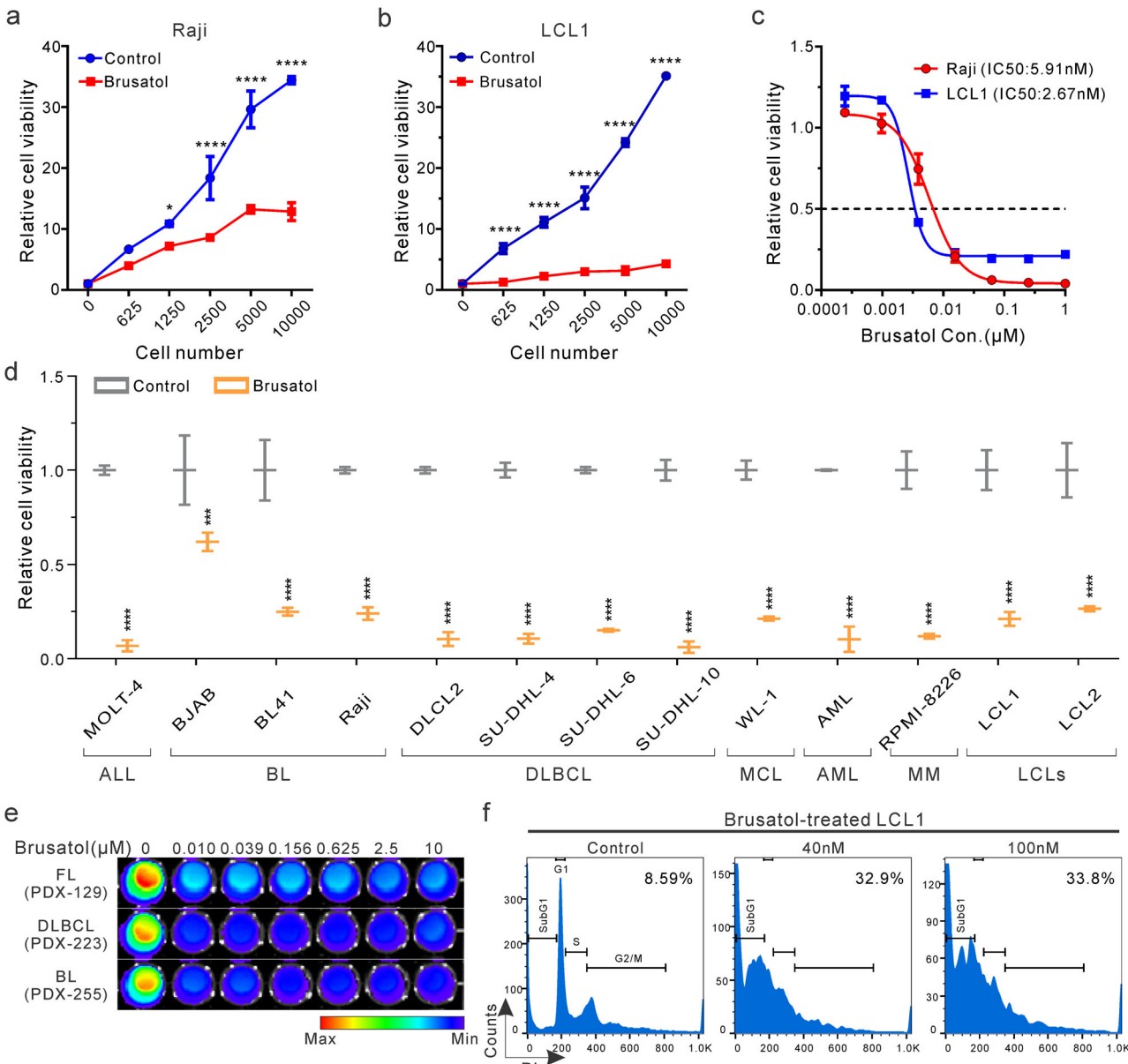

**Fig. 1 Brusatol inhibits viability of hematologic malignancy derived cells. a**, **b** A serial number of Raji (**a**) or LCL1 (**b**) cells were incubated with 100 nM Brusatol for 72 h. Cell viability was determined by detecting luminescence. Untreated cells were set as a negative control. Results are the mean ± standard error of the duplicates. *$p < 0.05$ and ****$p < 0.0001$ show the significant differences between Brusatol-treated cells and the control group. **c** 5000 Raji or LCL1 cells were incubated with a serial concentration of Brusatol for 72 h. Cell viability was determined by detecting luminescence and relative luminescence was shown by comparing to untreated samples. **d** 5000 indicated cells were untreated or treated with 100 nM Brusatol. After 72 h of incubation, cell viability was determined and shown. ALL, acute lymphoblastic leukemia; BL, Burkitt's lymphoma; DLBCL, diffuse large B-cell lymphoma; MCL, mantle cell lymphoma; AML, acute myeloid leukemia; MM, multiple myeloma; LCLs, lymphoblastoid cell lines. Results are the mean ± standard error of the duplicates. ***$p < 0.001$ and ****$p < 0.0001$ show the significant differences as compared to the corresponding control group. **e** Indicated PDXs were cultured in 96-well plates and treated with different concentrations of Brusatol for 72 h. Cell viability was monitored by testing luminescence. FL (PDX-129), B-cell Follicular lymphoma; DLBCL (PDX-223), Diffuse large B cell lymphoma; BL (PDX-255), Burkitt's lymphoma. **f** LCL1 cells were untreated or treated with Brusatol (40 nM, 100 nM) for 72 h. Then cells were fixed, stained, and analyzed with flow cytometry for cell cycle assay. The subG1 population in cells was labeled as the percentage.

Supplementary Data 1). These are also potential targets of Brusatol and are likely involved in regulation of downstream signaling to promote Brusatol-mediated inhibition. However, additional biochemical data will further provide evidence of the interaction with these candidates.

To definitively identify the direct targets of Brusatol, we synthesized a series of biotin-conjugated Brusatol derivatives to capture its targets using mass spectrometry (MS) analysis (Fig. 2d,

and Supplementary Fig. 2a). The structure-activity relationship (SAR) was determined to identify positions that allowed for the chemical conjugation of biotin without affecting its biochemical activity. Large linear esters at C-21 led to a reduction in potency, but the incorporation of nitric oxide-releasing groups at C-3 was better tolerated and retained activity[28,29]. Thus, the biotin-conjugated Brusatol analogs derived from positions C-3 and C-21 were examined to demonstrate their IC50 on multiple PDXs, and

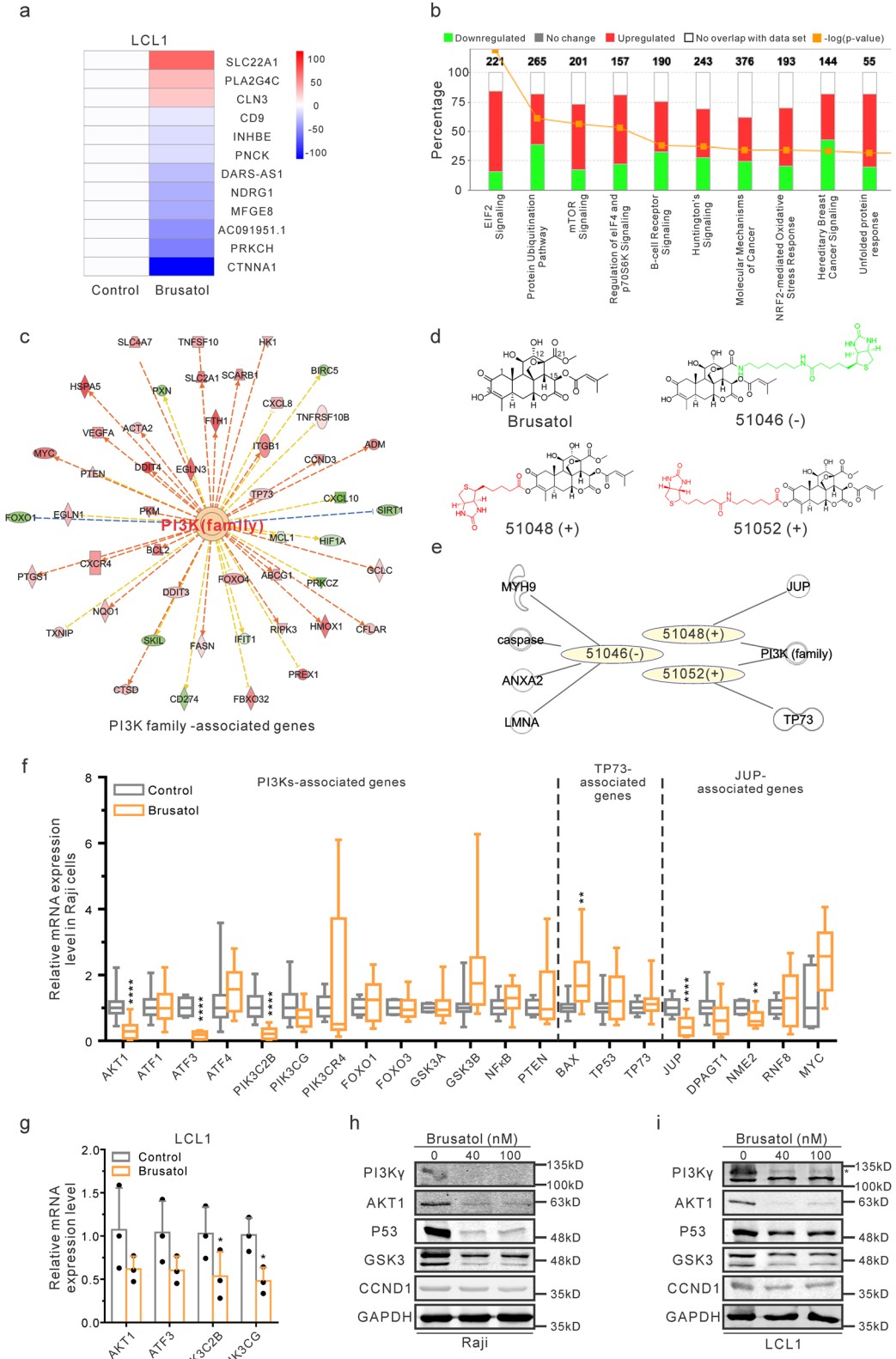

lymphoma cell lines (Supplementary Fig. 2). The results showed that the C-3 hydroxyl on Brusatol can be derivatized with a lipophilic tert-butoxy carbonyl (Boc) protected glycine or hydrophilic smaller amino acids such as beta-aminobutyric acid and proline, with minimal effects on its inhibitory ability, and was used to attach the biotin moiety. In contrast, a bigger isopropyl ester at C-21 resulted in a loss of activities (Supplementary Fig. 2b). Two additional amide derivatives from C-21 lost activity indicating that a large appendant is not tolerated at this position (Supplementary Fig. 2). Therefore, the active domain of Brusatol is associated with the C-21 position[29,30].

Next, we synthesized C-3-biotinylated Brusatol derivatives with different linkers (51048, 51052) and a C-21 biotin conjugate as a negative control (51046). These three biotin-conjugated Brusatol

**Fig. 2 Brusatol specifically targets the PI3K/AKT signaling pathway. a** RNA-Seq analysis was performed in Brusatol-treated LCL1 cells. The heatmap illustrates 12 genes with the greatest differences after Brusatol treatment. **b** Signaling pathway analysis of RNA-Seq data showed the percentages of regulated genes in these indicated pathways. The numbers above the bar charts described the total genes involved in these different pathways. **c** Upstream regulator analysis in the IPA program showed PI3K family members are the important upstream factors. **d** Chemical structures of the parent Brusatol and three biotin-conjugated Brusatol analogs were shown. The synthesized C-3-biotinylated Brusatol derivatives (51048, 51052) were set as the experimental samples (+), while a C-21 biotin conjugate (51046) was set as a negative control (−). **e** Correlation of mass spectrometry data and upstream regulator data were conducted using the IPA program. **f** Real-time PCR was performed in untreated or Brusatol-treated Raji cells to verify the mRNA expression of upstream regulators (PI3K family, TP73, JUP) associated genes. The results from three independent experiments were shown. Results are the mean ± standard error of the samples ($n = 3$). **$p < 0.01$ and ****$p < 0.0001$ show the significant differences as compared to the control group. **g** Real-time PCR was performed to determine the mRNA expression of PI3K/AKT signaling members in LCL1 cells after Brusatol treatment. Results are the mean ± standard error of the samples ($n = 3$). *$p < 0.05$ shows significant differences as compared to the control group. **h, i** Western blot analysis was conducted to show the protein expression of PI3K/AKT signaling factors in Raji (**h**) or LCL1 (**i**) cells after Brusatol treatment. The PI3Kγ band was labeled by an asterisk. Two bands of GSK3 were GSK3α (51kD) and GSK3β (47kD), respectively.

derivatives were selected for mass spectrometry (MS) experiments (Fig. 2d). Approximately 30 proteins were identified as potential targets for each derivative (Supplementary Data 2). To narrow down the candidates, the MS results were integrated with upstream regulators identified from the RNA-Seq analysis. Strikingly, PI3K family members were the only candidates identified with both 51048 and 51052 biotin-conjugated derivatives (Fig. 2e). Compound 51048 interacted with *PIK3CG* and *JUP* encoded proteins, while 51052 interacted with *PIK3C2B* and *TP73* encoded proteins (Supplementary Data 2). *PIK3CG* encodes PI3Kγ and *PIK3C2B* encodes PI3KC2β, both members of the PI3K family[31]. *JUP* encodes for the plakoglobin protein, known as junction plakoglobin or γ-catenin, which is important in acute myeloid leukemia (AML)[32]. The P73 protein is also widely studied in association with hematologic malignancies[33].

To validate these links of upstream regulators, their downstream associated genes were identified and the mRNA transcripts were monitored using Real-time PCR in Raji cells treated with Brusatol. The mRNA levels of PI3K family-associated AKT1, ATF3, PIK3C2B; *TP73*-associated BAX; and *JUP*-associated JUP and NME2 were significantly downregulated after Brusatol treatment (Fig. 2f). These may all be potential targets of Brusatol. However, we focused on the PI3K family members, PI3Kγ and PI3KC2β, as they were the only two proteins precipitated by both active biotin-conjugated Brusatol derivatives (Fig. 2e). Moreover, we wanted to determine if Brusatol regulated key members of the PI3Ks signaling pathways. The results demonstrated that AKT1, ATF3, PIK3CG, and PIK3C2B were downregulated at the mRNA levels in Brusatol-treated EBV-positive Raji and LCL1 cells (Fig. 2f, g). The levels of AKT1 and GSK3 proteins were also decreased on Brusatol treatment (Fig. 2h, i). This provided strong supporting evidence that Brusatol can target PI3K family members to perturb the PI3K/AKT signaling pathway.

**PI3K family is crucial for Brusatol-mediated inhibition**. To identify the mechanism by which Brusatol targets the PI3K/AKT pathway in hematologic malignancies, we screened several cell lines derived from different types of hematologic malignancies with varying levels of PI3Ks[24,25]. Notably, the IC50 showed that HL60 and K562 cell lines had reduced sensitivity to Brusatol treatment when compared to MOLM14 and SU-DHL-4 cell lines, which were about 10-fold more sensitive to Brusatol treatment at 72 h (Fig. 3a). Furthermore, Brusatol exhibited substantially lower inhibition of HL-60 and K562 cell lines compared to Raji and SU-DHL-4 cell lines (Fig. 3b). Specifically, at even 48 h post-treatment, Brusatol inhibited the viability of more than 80% of the Raji and SU-DHL-4 cells. However, the inhibitory effects were lower on HL-60 and K562 cell lines (Fig. 3b). Additionally, cell cycle analyses demonstrated that HL-60 and K562 cells were less

sensitive than Raji and SU-DHL-4 cells. After 48 h of treatment with Brusatol, 68.4% of Raji cells and 67.9% of SU-DHL-4 cells were in the sub-G1 phase. However, only 7.63% of HL-60 cells and 3.19% of K562 cells were in a similar phase (Fig. 3c). Therefore, we selected HL-60 and K562 as our Brusatol-less sensitive set of cells, and Raji and SU-DHL-4 as the Brusatol-more sensitive set of cells for further examination.

To compare the expression of PI3K family members and related molecules, we determined their endogenous levels in the two sets of cell lines (Brusatol-less or -more sensitive cells). We demonstrated that cell lines with reduced levels of PI3Kγ and PI3KC2β were shown to be less sensitive to Brusatol treatment (Fig. 3d). Interestingly, along with PI3Kγ, PI3Kδ, and P53 were also highly expressed in the two Brusatol-more sensitive cell lines (Fig. 3d). Further, Brusatol-less sensitive or Brusatol-more sensitive sets were incubated with increasing concentrations (0, 50 nM, 100 nM) of Brusatol at three different time points (0, 12 h, 24 h). The results showed that the protein expression of AKT1, GSK3, and mTOR was substantially reduced in Raji and SU-DHL-4 cells (Brusatol-more sensitive), and minimally affected in HL-60 and K562 cells (Brusatol-less sensitive) (Fig. 3e). This provides a possible explanation of why Raji and SU-DHL-4 cells were more sensitive and HL-60 and K562 showed less sensitivity in our cell cycle and cell viability assays. Therefore, we propose that Brusatol can specifically inhibit the PI3K/AKT/GSK3/mTOR signaling pathway.

**Brusatol can directly target the PI3Kγ isoform**. To further explore the effects of these compounds on other cancers, three patient-derived nasopharyngeal carcinoma (NPC) cell lines (C17, NPC43, and NPC53) were evaluated[34,35]. Cell viability assays demonstrated that these NPC cell lines are less sensitive to Brusatol when compared to the SU-DHL-4 cell line by IC50 (Fig. 4a). Notably, these NPC cells expressed relatively high levels of PI3KC2β, but PI3Kγ was minimally detected (Fig. 4b). Therefore, these results provide additional evidence that supports PI3Kγ as a potential target of Brusatol and is a critical potential target for cancer therapies[4,6,36].

Next, we also performed biotin-conjugated pull-down assays to examine the direct association of Brusatol with the PI3Kγ isoform. To perform the competitive binding assays, cell lysates from SU-DHL-4 cells were incubated with the biotin-conjugated Brusatol derivative (51048) alone or together with the parent Brusatol. The results showed that the binding activity of the Brusatol derivative (51048) with PI3Kγ was relatively strong but was reduced in the presence of the parent Brusatol compound (Fig. 4c). Additionally, the biotin-conjugated Brusatol derivative specifically interacted with isoform PI3Kγ, but had little or no detectable levels of AKT1 and GAPDH (Fig. 4c). Furthermore, GST-tagged PI3Kγ protein was purified to determine its

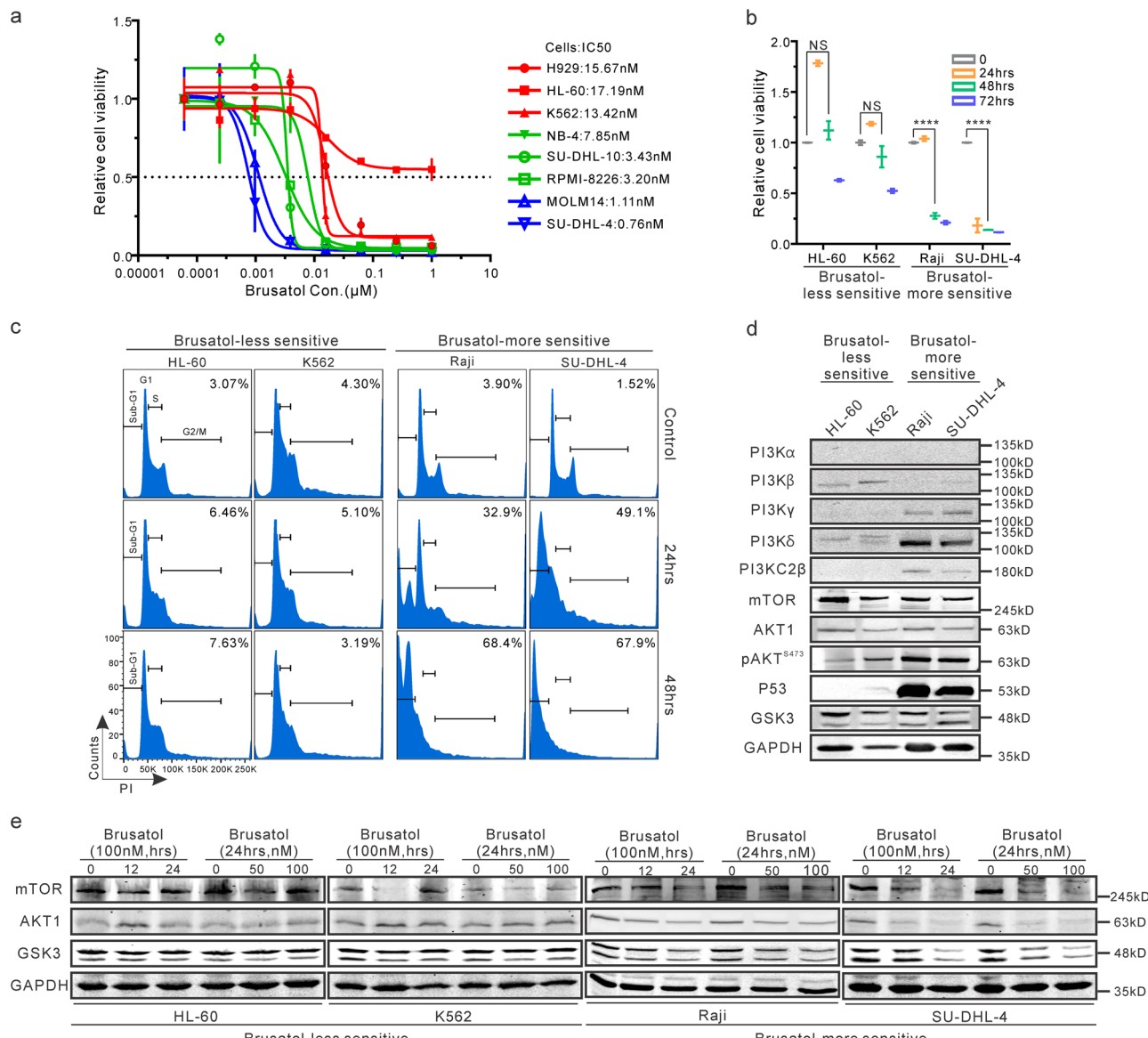

**Fig. 3 PI3K family is crucial for Brusatol-mediated inhibition. a** IC50 assays of Brusatol were determined in the indicated cancer cells. All of the cells were treated for 72 h. **b** The indicated Brusatol-less sensitive or -more sensitive cells were treated with 100 nM Brusatol for indicated periods (0, 24 h, 48 h, 72 h), and cell viability was examined by detecting luminescence. Results are the mean ± standard error of the duplicates. ****$p < 0.0001$ shows the significant differences between the indicated groups. NS, not significant. **c** Brusatol-less sensitive or -more sensitive cells were treated with 100 nM Brusatol for different periods (0, 24 h, 48 h). Then cells were fixed and stained with PI. Flow cytometry assays were performed to determine cell cycle progression. The subG1 population in cells was labeled as the percentage. **d** The endogenous expression of PI3K/AKT signaling protein in these cell lines, including sets of Brusatol-less sensitive and -more sensitive cells, was detected with western blot. **e** The described cells were incubated with Brusatol (0, 50 nM, 100 nM) for different periods (0, 12 h, 24 h). Western blot analysis was performed to determine the expression of PI3K/AKT regulated signaling proteins in Brusatol-less sensitive cells set (HL-60, K562) and Brusatol-more sensitive cells set (Raji, SU-DHL-4).

association with the biotin-conjugated Brusatol derivative (51052) similar to 51048 at the C-3 position for conjugation in vitro. The pull-down assays demonstrated that GST-tagged PI3Kγ formed a complex with the biotin-conjugated Brusatol derivative (51052) in vitro (Fig. 4d).

To further support our findings, we designed sgRNA to target the exon of the *PIK3CG* gene (Fig. 4e), and generated knock-out (KO) Raji cell lines using CRISPR/Cas9 system (Fig. 4f). Notably, the knock-out of the *PIK3CG* gene in Raji cells did not affect expression of other downstream molecules of the PI3K/AKT pathway (Fig. 4g). However, AKT1 level showed no obvious decrease in the knock-out cell line with Brusatol treatment, suggesting that crucial downstream targets of PI3Kγ protein were

specifically regulated (Fig. 4h). GSK3, mTOR, and P53 also showed less reduction in levels after Brusatol treatment in the knock-out cells (Fig. 4h). These results demonstrated that Brusatol can regulate the PI3K/AKT signaling pathway by targeting PI3Kγ to inhibit the viability of hematologic malignancies derived cancer cells.

**Development of novel Brusatol analogs with great efficacy.** To further improve the potential of Brusatol as a therapeutic agent, we synthesized a series of novel Brusatol analogs and evaluated their activity to identify candidates with increased bioactivity and minimal toxicity. 10 selected analogs from around 70 of our strategically

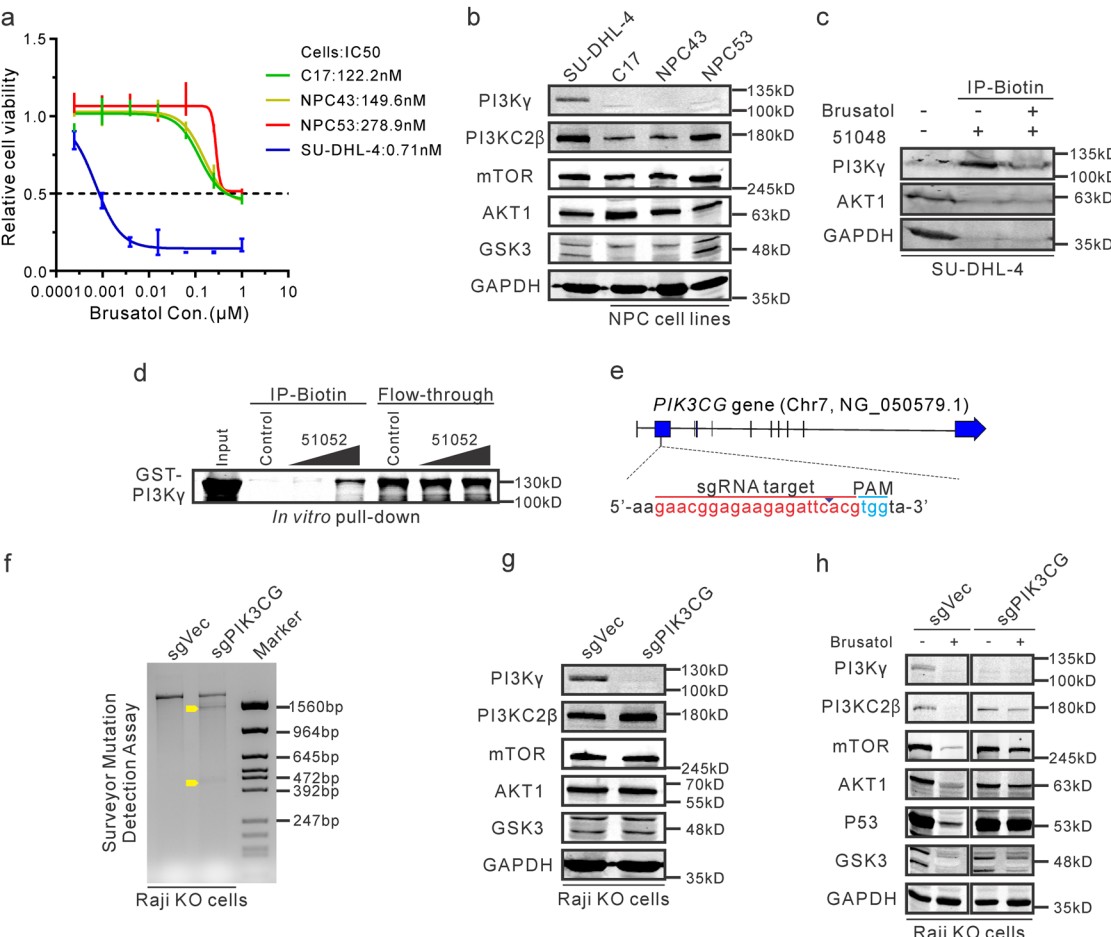

**Fig. 4 Brusatol can directly target the PI3Kγ isoform. a** IC50 of Brusatol was determined in these NPC cell lines (C17, NPC43, and NPC53) and Brusatol-sensitive SU-DHL-4 cells at 72 h post-treatment. **b** These cell lines were harvested and detected the endogenous expressions of indicated proteins with western blot analysis. **c** The lysates of SU-DHL-4 cells were incubated with biotin-conjugated 51048 compound together in the absence or presence of Brusatol, then the immunoprecipitated complex was detected with western blot. **d** In vitro expressed and purified GST-tagged PI3Kγ protein was incubated with 0, 300 μM, or 500 μM biotin-conjugated 51052 compound as well as Dynabeads M-280 Streptavidin. Then the binding protein was examined with western blot analysis. **e** A schematic diagram shows the target site of *PIK3CG* genes using the CRISPR/Cas9 system. **f** A Surveyor mutation detection assay was performed to verify whether the *PIK3CG* gene was mutated in knock-out (KO) Raji cells. The control cell line (sgVec) by transfecting empty plasmids were used as control. The yellow arrows indicated the truncated fragments. **g** The indicated proteins were detected in knock-out Raji cells by western blot analysis. **h** Knock-out Raji cells were untreated or treated with 100 nM of Brusatol for 72 h, then cells were harvested and determined the expressions of PI3K/AKT associated proteins with western blot.

generated compounds maintained considerable inhibitory effects on Raji cells compared to the parent Brusatol (Supplementary Fig. 3a–c). To further investigate the effects of these active compounds, the panel of hematologic malignancy derived cell lines were used. One inactive compound #1 was set as a negative control (Supplementary Fig. 3d). The results showed that four representative analogs, #14, #15, #26, and #31 were able to significantly inhibit cell viability of all 13 cell lines representing a range of hematologic malignancies (Fig. 5a). Interestingly, the selected compounds that remained active were all modified at the C-3 position (Fig. 5b), supporting our conclusion above that the active domain of Brusatol is not likely to be associated with this modification (Fig. 2d, and Supplementary Fig. 2). These novel Brusatol analogs showed comparable efficacy in MOLT-4, SU-DHL-4, SU-DHL-10, and RPMI-8226 cells as more than 90% of viable cells were inhibited after treatment (Fig. 5a). Expectedly, the inactive compound #1 was unable to inhibit the growth of these cells in vitro (Fig. 5a). Further, the IC50 showed that these promising candidates maintained effective inhibitory effects comparable to what was previously seen above with the parent Brusatol (Fig. 5c).

To examine the anti-cancer effects of these Brusatol analogs in vivo, 6-week-old male NOD.CB17-Prkdc[scid]/J (NOD/SCID) mice were used as the human-in-mouse xenograft-transplantation model by injecting MOLM14 cells, which were also sensitive to Brusatol and was substantially inhibited in our cell viability and cell cycle assays (Fig. 3a, and Supplementary Fig. 3e, f). MOLM14 cells also highly express PI3Kγ but not PI3KC2β (Supplementary Fig. 3g). Moreover, the MOLM14 mouse model is a straightforward and rapid in vivo assay for preclinical studies[37]. Here, MOLM14 cells were injected subcutaneously, and when xenografts reached 100 mm³ the compounds (Brusatol, #14, #15, #26, #31) or PBS buffer were injected intraperitoneally three times weekly. After 2 weeks of treatment, the mice were euthanized and tumors were excised (Supplementary Fig. 3h). The tumor size of the treated mice clearly showed that analogs #15 and #26 exhibited a strong and rapid inhibitory response compared to analogs #14 and #31 tested (Fig. 5d, e). The monitored weight of the mice did not show any obvious changes (Supplementary Fig. 3i). Therefore, these results show that these new analogs of Brusatol had significantly comparable bioactivity with the parent Brusatol.

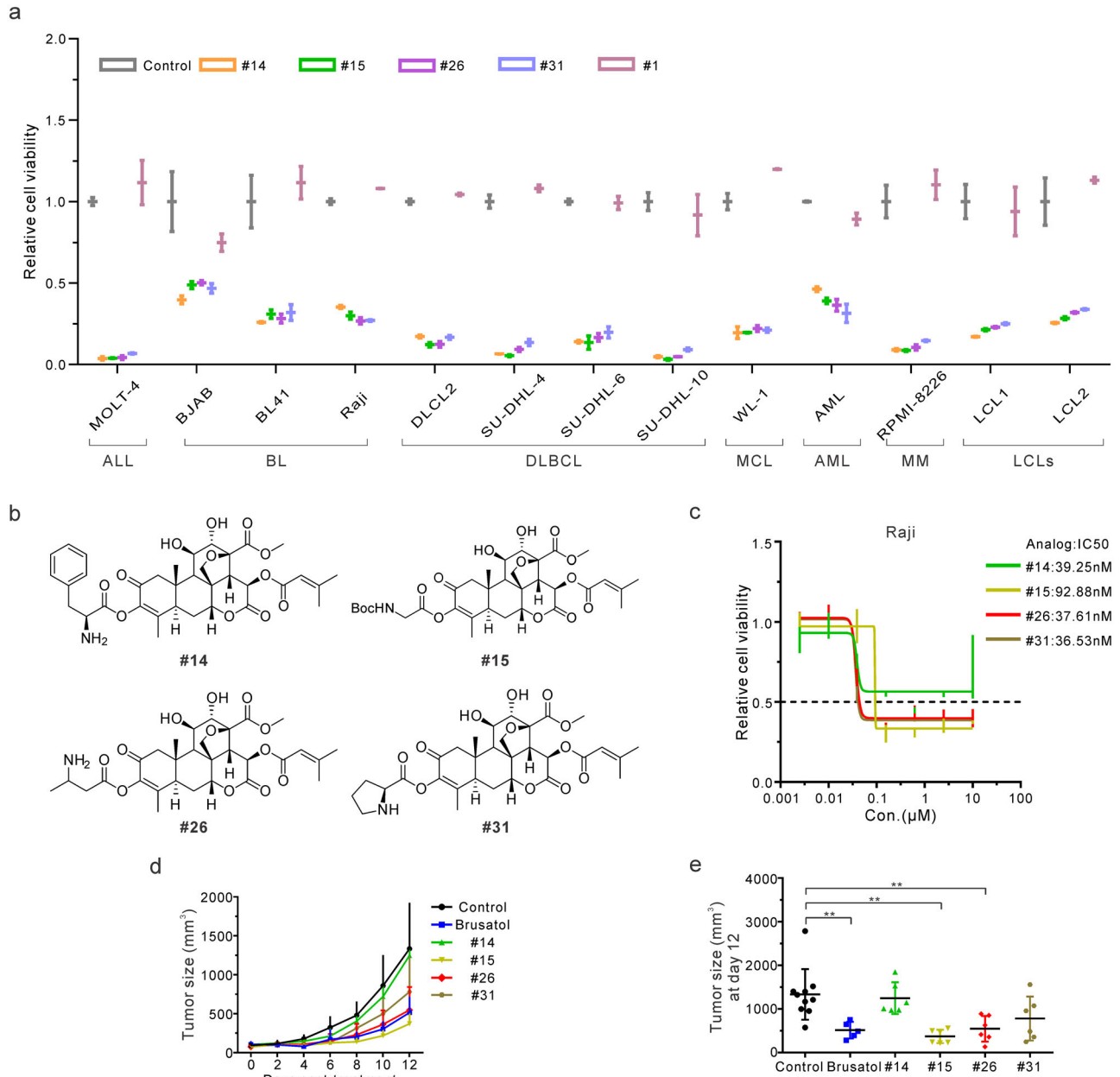

**Fig. 5 Development of novel Brusatol analogs with great efficacy. a** Multiple hematopoietic malignant cell lines were untreated or treated with 100 nM of #14, #15, #26, #31, or #1 compound. Cell viability was determined after 72 h by detecting luminescence. ALL, acute lymphoblastic leukemia; BL, Burkitt's lymphoma; DLBCL, diffuse large B-cell lymphoma; MCL, mantle cell lymphoma; AML, acute myeloid leukemia; MM, multiple myeloma; LCLs, lymphoblastoid cell lines. Results are the mean ± standard error of the duplicates. **b** Chemical structures of these tested Brusatol analogs were shown. **c** Raji cells were treated with different concentrations of these analogs for 72 h to determine their IC50 using cell viability assays. **d** Tumor sizes were monitored in the mice after treatment of the indicated compounds. Results are the mean ± standard error of the mice (n > 5). **e** Tumor size at 12-day post-treatment with different analogs was highlighted. Results are the mean ± standard error of the mice (n > 5). **p < 0.01 shows significant differences as compared to the control group.

**One novel Brusatol analog is a potential PI3K inhibitor**. A crucial limitation in the development of Brusatol as a therapeutic agent was linked to its associated toxicities[38]. To examine the potential cytotoxicity of these compounds, human PBMC, T-cells, and B-cells were treated with these compounds (Brusatol, #15, and #26), three FDA approved inhibitors (Copanlisib, Duvelisib, and Idelalisib), and IPI-549 in clinical trials. Importantly, our analogs did not show much higher cytotoxicity in these normal human cell lines when compared with the approved PI3K inhibitors (Fig. 6a–c). Furthermore, cell viability assessed in Brusatol-sensitive cells, including SU-DHL-4, MOLM14, and Raji, showed

that inhibition exhibited by the two analogs were comparable to the pan-PI3K inhibitor Copanlisib, and had greater efficacy compared to the other PI3K inhibitors (Fig. 6d–f). These results provide strong evidence that Brusatol and the novel analogs have the potential for clinical use as PI3K inhibitors. It should be noted that overexpressed PI3Kδ in the Brusatol-sensitive cells may also be a potential target as it was also upregulated but would require further investigation (Fig. 3d).

To investigate whether the two novel analogs #15 and #26 also specifically targeted the PI3K/AKT signaling pathway, the Brusatol-less sensitive HL-60 cells, and Brusatol-more sensitive

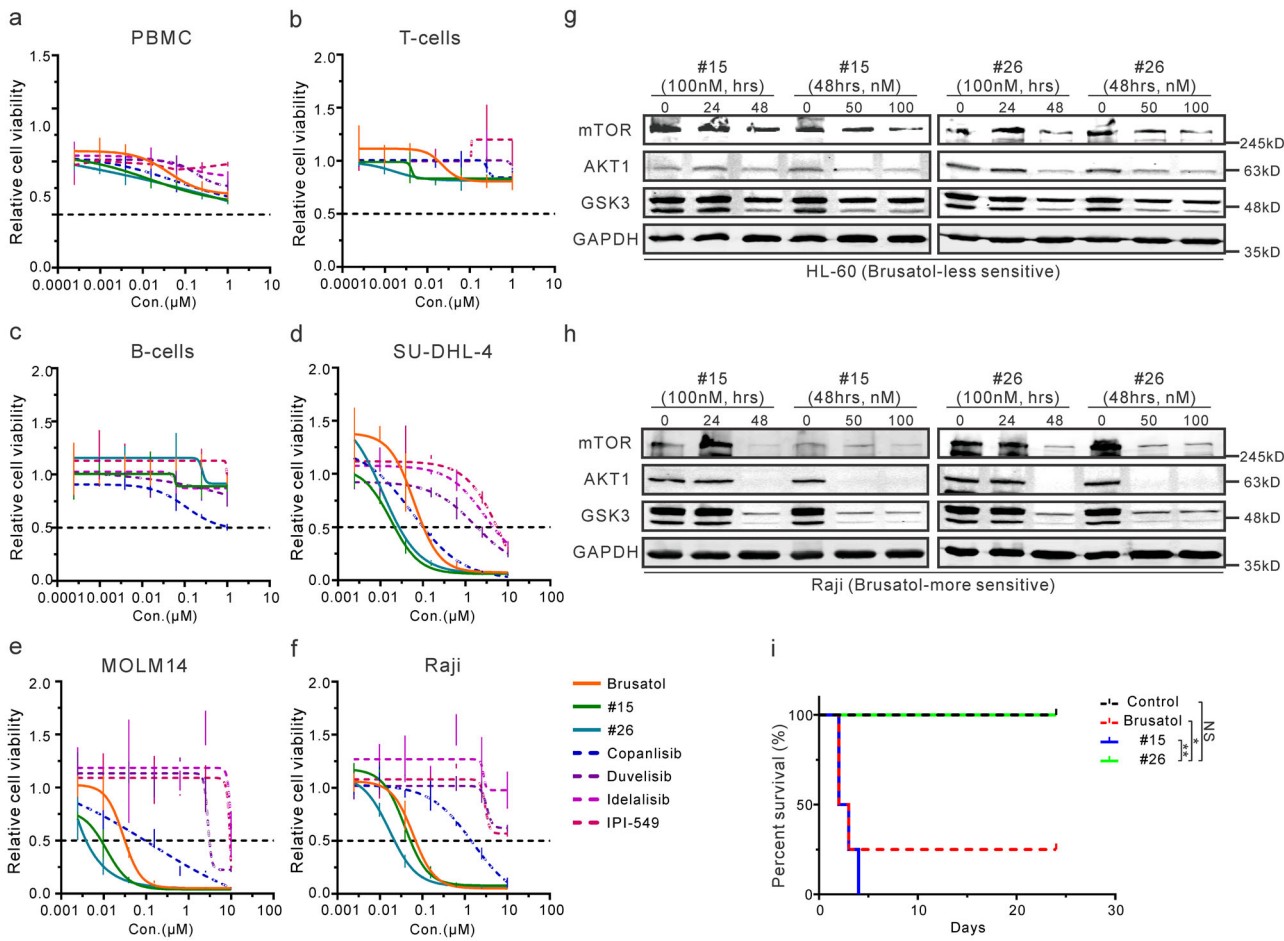

**Fig. 6 One novel Brusatol analog is a potential PI3K inhibitor. a–c** IC50 assays were performed by treating PBMC (**a**), normal T-cells (**b**), or normal B-cells (**c**) with Brusatol analogs and other PI3K inhibitors. PBMC and normal T-cells were treated for 72 h, and normal B-cells were treated for 24 h. **d–f** IC50 of Brusatol analogs and PI3K inhibitors was investigated in SU-DHL-4 (**d**), MOLM14 (**e**), and Raji (**f**) cells. All of these cells were treated for 72 h. **g–h** HL-60 (**g**) and Raji (**h**) cells were incubated with analog #15 or #26 (0, 50 nM, 100 nM) for indicated times (0, 12 h, 24 h). Western blot analysis was performed to determine the expression of PI3K/AKT regulated signaling proteins. **i** NOD/SCID mice (4 per group) were intraperitoneally injected 10 mg/kg of indicated compounds (Brusatol, #15, and #26) every other day. The survival rate was monitored and demonstrated to show their toxicity in vivo. Results are the survival rate of the mice (n = 4). *$p < 0.05$ and **$p < 0.01$ show the significant differences between the indicated groups. NS, not significant.

Raji cells were treated with these analogs for different periods. Western blot analyses showed that the developed analogs were largely similar to Brusatol in the suppression of AKT1, GSK3, and mTOR levels in Raji, but not HL-60 cells (Fig. 6g, h). These findings demonstrated that these analogs also targeted the PI3K/AKT signaling pathway. However, #26 analog inhibited both PIK3CG and GSK3B mRNA expression in vivo, which was different from the inhibitory effects of Brusatol and #15 (Supplementary Fig. 3j). Therefore, the potential anti-cancer mechanisms of the analog need further investigation.

We then tested the toxicity of these compounds in vivo. The activate compounds (Brusatol, #15, and #26) were intraperitoneally injected with 10 mg/kg every other day. The mice were monitored and euthanized after 24 days of treatment. Brusatol and analog #15 were obviously lethal for the majority of mice within the first 4 days. However, analog #26 did not show any obvious signs of toxicity even after more than 3 weeks of treatment (Fig. 6i). These results demonstrated that #26 analog has a significantly reduced toxicity compared to Brusatol and analog #15. Therefore, the novel analog (#26) had remarkable inhibitory effects combined with less or minimal toxicity, demonstrating strong potential for development as a small-molecule therapeutic for the treatment of EBV-positive and other hematologic malignancies.

## Discussion

The quassinoid family member Brusatol was previously shown to exhibit anti-cancer activity on human leukemia cells[10,11]. Our studies now show that Brusatol can also inhibit EBV-transformed or positive cells as well as multiple hematologic malignancies derived cell lines. Furthermore, it showed effective inhibition of hematologic PDX cells demonstrating its potential as a therapeutic for treating these cancers. Brusatol was previously recognized as an inhibitor of Nrf2 and can suppress protein synthesis[12]. However, the mechanism of its anti-cancer activity has not been fully explored. It is also unclear why Brusatol inhibits eukaryotic protein synthesis with minimal hematologic toxicity as seen in normal PBMCs. Our studies now show that Brusatol had inhibitory activity on multiple cell lines, suggesting that the complex mechanism of Brusatol's anti-cancer activity may involve multiple direct targets or cellular processes.

In this study, we identified a direct target of Brusatol and showed that its associated signaling pathway was dysregulated using mass spectrometry and RNA-Seq analyses. The data demonstrated that Brusatol effectively disturbed PI3K/AKT signaling pathway, critical for many downstream activities, including the Nrf2 signaling pathway, mTOR-associated protein regulation, and proteins associated with cell cycle or cell

apoptosis[39–41]. Inhibition of PI3K/AKT signaling by Brusatol was validated using two sets of cell lines with different sensitivities to the compound. Furthermore, dysregulation of PI3K is an important hallmark of many human cancers[1], and targeting PI3K family members is a strategic approach for the treatment of these cancers[2,4,6]. Our in vitro assays confirmed PI3Kγ as a direct target of Brusatol. PI3Kγ is a molecular switch involved in controlling immune stimulation and suppression of inflammation and cancer. This suggests that Brusatol may have potential synergistic activities when combined with T-cell-targeted therapy through disruption of PI3Kγ activity[36,42,43]. However, our experiments also demonstrated that these Brusatol-sensitive cells were not resistant after knock-out of only PI3Kγ isoform. This suggests that Brusatol may also target other critical cellular factors, which may have compensatory functions to the PI3Kγ protein, and are driving the survival of these knock-out cells. This is highly likely as we did identify other potential targets from our mass spectrometry results. These will be explored in future studies.

To explore the potential anti-cancer mechanisms of Brusatol, we took a broader view of this natural compound. Based on the previous results, P53 and P73 proteins are other potential targets in addition to PI3Kγ protein (Fig. 2e, Fig. 4h, and Supplementary Data 1). P73 was demonstrated to interact with compound 51052 (biotin-conjugated Brusatol analog) as determined by our mass spectrometry results, and its regulatory downstream *BAX* gene was also upregulated after Brusatol treatment (Fig. 2f). *BAX*-encoded BCL2L4 protein, also known as Bcl-2-like protein 4, belongs to the Bcl-2 gene family and functions as an apoptotic activator[44,45]. P73-associated apoptosis is mediated by PUMA and further induced BCL2L4 mitochondrial translocation and Cytochrome C release[46]. Besides, *BAX* can also be regulated by the tumor suppressor P53[47–49]. The *TP53* gene was also identified in our upstream regulatory pathway analysis, so it was reasonable to infer that P53 and P73 may be potential targets of Brusatol because of their high homology and overlapping functions[50]. *JUP*-encoded γ-catenin also interacted with the other active compound 51048 (biotin-conjugated Brusatol analog) (Fig. 2e). γ-catenin regulates NME2 mRNA expression which was down-regulated in Brusatol-treated cells (Fig. 2f). Furthermore, it can also interact with the metastasis suppressor Nm23-H2 encoded by the *NME2* gene and promotes its activities[51]. However, a detailed mechanism would need further investigation.

In this study, our results demonstrated that Brusatol induced cell death of a range of human hematologic malignant cells by targeting PI3Kγ and suppressing the PI3K/AKT signaling pathway. These included EBV-positive Burkitt's lymphoma cells which also showed increased levels of PI3K pathway activation[52], as well as many other refractory lymphoma and leukemia that are frequently relapsed in the clinic. Additionally, EBV latent antigens LMP1, LMP2A, as well as KSHV-encoded transmembrane glycoprotein K1, G protein-coupled receptor (vGPCR), viral IL-6 (vIL-6) can constitutively activate the PI3K/AKT pathway to promote virus-mediated tumorigenesis[53–57]. This strongly suggests that Brusatol can be further developed as a precision therapeutics for treatment of tumor-virus induced hematologic diseases by targeting activated PI3K/AKT pathway.

The Quassinoid family includes more than 150 compounds that are extracted from the Simaroubaceae plant family[9]. Many quassinoid members show anti-inflammatory and anti-cancer activities, but the exact mechanism of their actions requires further investigation[9,11,58]. Several studies showed that Brusatol induced cell differentiation was associated with MYC down-regulation, and NF-κB activation[24,25]. Additionally, the toxicity of quassinoid members has been a major obstacle limiting their applications as anti-cancer drugs in clinical trials[59,60]. However, modifications of these natural compounds have potential value

for the development of therapeutic agents for targeted cancer treatment.

To develop Brusatol as a therapeutic drug for treating hematologic malignancies in early phase clinical trials, we strategically synthesized a series of Brusatol analogs and examined their activity on a broad range of hematologic malignancy derived cell lines. These analogs modified at the C-3 position demonstrated considerable inhibition of these cancer cells. Moreover, when compared to the commercially available PI3K inhibitors, these analogs also exhibited comparable cytotoxicity. Notably, analogs #15 and #26 exhibited similar levels of inhibition to Brusatol in our xenografts model in vivo. We cannot conclude at this time that their anti-cancer mechanisms were similar to the parent Brusatol. However, they both substantially inhibited the PI3K/AKT pathway with specific effects on cells with increased PI3Kγ levels. Furthermore, analog #26 showed remarkably reduced toxicity in vivo than Brusatol and analog #15. Our data thus demonstrate that the novel analog #26 forms the basis of a platform for the development of targeted small-molecule therapeutics for treatment of EBV-positive, and hematologic malignancies in early phase clinical trials with comparative efficacy to current PI3K inhibitors in terms of their potency, but with specificity for PI3Kγ. This adds to their precision in targeting hematologic malignancies with dysregulated PI3Kγ levels.

## Methods

**Ethical statement**. Human peripheral blood mononuclear cells (PBMCs), normal T-cells and B-cells were provided by the University of Pennsylvania Human Immunology Core (HIC), and Patient-derived xenografts (PDXs) were obtained from Fox Chase Cancer Center (FCCC). These samples were from different unidentified and healthy donors with written, informed consent. All the procedures were approved by the Institutional Review Board (IRB) and conducted according to the declarations of Helsinki protocols.

The in vivo experiments were performed in a preclinical mouse model and conducted by following the Office of Laboratory Animal Welfare (OLAW) guidelines provided by National Institute of Health (NIH, USA) under the supervision and management of the University of Pennsylvania Institutional Animal Care and Use Committee (IACUC) with approved protocol #804549. The number of independent experiments or replicates are indicated in the figure legends.

**Cell lines, antibodies, primers, and compounds**. Cell lines, antibodies, primers, and compounds used in this study are summarized in Supplementary Table.

**General methods for chemistry**. All reactions were carried out under argon atmosphere. Preparative high-performance liquid chromatography (HPLC) was performed using a Gilson 331 and 332 pumps with a UV/VIS-155 detector and GX-271 liquid handler. Column was Phenomenex Luna LC Column (5 μm C18 100 Å, 150 × 21.2 mm). [1]H NMR spectra were recorded on a 300 MHz INOVA VARIAN spectrometer. Chemical shift values are given in ppm and referred as the internal standard to TMS (tetramethylsilane). The coupling constants (*J*) are reported in Hertz (Hz). Mass Spectra were obtained on an Agilent 6120 mass spectrometer with electrospray ionization source (1200 Aligent LC-MS spectrometer, Positive). Mobile phase flow was 1.0 mL/min with a 3.0 min gradient from 20% aqueous media (0.1% formic acid) to 95% $CH_3CN$ (0.1% formic acid) and a 9.0 min total acquisition time. All the tested compounds possess a purity of at least 95%, which was determined by LC/MS Data recorded using an Agilent 1200 liquid chromatography and Agilent 6120 mass spectrometer, and further supported by clean NMR spectra. Generation of these tested compounds was described in the Supplementary Method.

**Cell viability assay**. 5000 cells as indicated were cultured in 96-well plate with 100 μl of media. On the next day, Brusatol or associated analogs were added once to the cells without media change. After treatment, the cells were incubated with 100 μl buffer for 10 min at room temperature following the instructions of CellTiter-Glo kit (Promega, Madison, WI). Luminescence was detected using a multi-mode reader Cytation 5 (BioTek, Winooski, VT). Brusatol or its analogs are dissolved in 100% ethanol and diluted in 1 × PBS for treatment. Two independent experiments done in duplicates were performed.

**Cell cycle assay**. Five million cells were harvested, fixed with 80% ethanol for 2 h or overnight at −20 °C, then washed with PBS and incubated with Propidium iodide (PI) staining buffer (0.5 mg/ml propidium iodide in PBS with 50 μg/ml

RNase A) for 30 min at room temperature. The stained cells were resuspended in PBS and analyzed on a FACSCalibur system (Becton Dickinson, San Jose, CA, USA) using FlowJo software (TreeStar, San Carlos, CA, USA).

**Mass spectrometry (MS)**. Two hundred million Raji cells were harvested and lysed with RIPA buffer [1% Nonidet P-40 (NP-40), 10 mM Tris (pH8.0), 2 mM EDTA, 150 mM NaCl, supplemented with protease inhibitors (1 mM phenylmethylsulphonyl fluoride (PMSF), 1 µg/ml each Aprotinin, Pepstatin, and Leupeptin)]. Twenty milligram lysates were combined with magnetic Streptavidin beads as well as biotin-conjugated compounds 51046, 51048, or 51052, respectively. The complexes were incubated overnight with rotation at 4 °C. The immunoprecipitated samples were washed with RIPA three times and resolved on an SDS-PAGE gel.

The following Mass Spectrometry proteomics resources and services are provided by the Quantitative Proteomics Resource Core at School of Medicine at the University of Pennsylvania. Briefly, the peptides were extracted from the gel band and analyzed with an Orbitrap Fusion (Thermo Fisher Scientific, San Jose, CA, USA) attached to an EasyLC 1000 system (Thermo Fisher Scientific, San Jose, CA, USA) at 400 nL/min. The raw MS data were processed using MaxQuant (Version 1.5.3.30)[61], and searched with the UniProt human database. Then the target-decoy approach was used to filter the search results, in which the false discovery rate was less than 1% at the peptide and protein level[62].

**RNA-Seq and data analysis**. LCL1 cells were untreated or treated with Brusatol for 72 h, then total RNAs from these cells were extracted with Trizol (Invitrogen, Inc., Carlsbad, CA) as previously described[63]. The cDNA library was prepared with the commercial Illumina library preparation kits (TruSeq Stranded RNA LT Ribo-Zero H/M/R Kit) according to the manufacturer's protocols and sequenced with an Illumina HiSeq2000 instrument (Washington University Genome Sequencing Center). Quality check was performed on the raw RNA-Seq reads (FastQC, https://www.bioinformatics.babraham.ac.uk/projects/fastqc/), and the adapters were cut (Trim Galore, https://www.bioinformatics.babraham.ac.uk/projects/trim_galore/) and mapped with hg38 genome (HISAT2)[64]. Differentially expressed transcripts were defined with Cuffdiff tools[65] and visualized using R software (https://www.r-project.org/). Signaling pathway analysis was conducted with the Ingenuity Pathway Analysis program (QIAGEN Inc., https://www.qiagenbioinformatics.com/products/ingenuitypathway-analysis)[27].

**Real-time quantitative PCR**. Real-time PCR experiments were performed as previously described[63]. The experiments were performed in triplicate.

**Proteins expression and purification**. PIK3CG cDNA (Plasmid #20574) was purchased from Addgene and cloned into a pGEX-6P-1 vector (GE Healthcare, Madison, WI, USA) with SalI and NotI (NEB, USA). This GST-tagged PIK3CG plasmid was transformed into *E. coli* BL21 (DE3)-competent cells (Life Technologies, Carlsbad, CA, USA) for in vitro expression according to Molecular Cloning (Third Edition).

**Pull-down assay**. Thirty microgram of purified GST-tagged proteins were pre-cleared with Dynabeads M-280 Streptavidin (Invitrogen, Carlsbad, CA, USA). The supernatants were collected and incubated with Dynabeads M-280 Streptavidin and 0, 300 µM, or 500 µM of biotin-conjugated Brusatol derivatives (51052) overnight at 4 °C. For the competitive binding assays, cell lysates from SU-DHL-4 cells were incubated with the biotin-conjugated Brusatol derivative (51048) alone or together with Brusatol as well as Dynabeads M-280 Streptavidin. The complexes were harvested and washed with PBS. The collected beads were mixed with SDS loading buffer for further analysis.

**Western blot**. Western blot analyses were performed as previously described[63]. Briefly, total cell lysates were separated by SDS-PAGE gel and transferred to nitrocellulose membrane. The membranes were blocked with 5% non-fat milk, probed with specific antibodies, and visualized using the Odyssey scanner (LiCor Inc., Lincoln, NE).

**Generation of CRISPR knock-out cell lines**. Oligos were obtained from Integrated DNA Technologies (IDT; Coralville, IA, USA) and cloned into the lentiviral vector lentiCRISPR v2 (Addgene #52961)[66]. Lentivirus production and transduction have been described previously[63]. Here, Raji cells were transduced with a lentivirus harboring lentiCRISPR v2 vector or a construct containing the guide sequence (GAACGGAGAAGAGATTCACG) against *PIK3CG* gene. At 48 h post-transduction, cells were selected with puromycin and harvested to determine PIK3CG expression with western blot.

**Surveyor mutation detection assay**. Genomic DNAs were extracted with DNeasy Blood & Tissue Kits (Qiagen, Valencia, CA, USA). PCR amplicons of target regions were analyzed for the mismatch mutations using Surveyor Mutation Detection Kit (Integrated DNA Technologies) according to the manufacturer's instructions.

**In vivo tumorigenic assays using xenografts**. Six-week-old male NOD.CB17-Prkdc^scid/J (NOD/SCID) mice (Jackson Labs, Bar Harbor, ME, USA) were used as the human-in-mouse xeno-transplantation model. MOLM14 cells ($1 \times 10^7$ cells) were injected into the subcutaneous space on the left flank of the mice. Once xenografts reached a size of 100 mm$^3$, Brusatol and analogs (#14, #15, #26, #31) at 2 mg/kg body weight or PBS were injected intraperitoneally three times per week. Body weight and tumor size were monitored before every injection. After treatment for 2 weeks, the mice were euthanized by $CO_2$ inhalation and the tumors were carefully excised. The tumor volume (mm$^3$) was measured and calculated using the formula $(a \times b \times b)/2$ ($a$, the largest diameter of two measurements; $b$, the smallest diameter of two measurements).

**In vivo toxicity test**. To determine the toxicity of Brusatol, #15 and #26 compounds, 6-week-old male NOD.CB17-Prkdc^scid/J (NOD/SCID) mice (Jackson Labs, Bar Harbor, ME, USA; 4 mice per group) were intraperitoneally injected with a maximum dose of 10 mg/kg of Brusatol, #15 or #26 compound every other day (three times per week). The survival rate was recorded to monitor the tolerance of the compounds in these mice.

**Statistics and reproducibility**. GraphPad Prism was used for statistical analysis. The mean values with standard deviation (SD) were presented in this study when appropriate. The significance of differences was calculated by performing a 2-tailed student's t-test. The P-value of < 0.05 was considered as statistically significant in our results. NS, not significant; *P-value < 0.05; **P-value < 0.01; ***P-value < 0.001; ****P-value < 0.0001.

**Reporting summary**. Further information on research design is available in the Nature Research Reporting Summary linked to this article.

## Data availability

All data associated with this study are available in the main text or the supplementary information. RNA-Seq data are available at the NCBI Gene Expression Omnibus (GEO) with accession number GSE130728. Mass spectrometry data are available at the Mass Spectrometry Interactive Virtual Environment (MassIVE) with accession number MSV000085067.

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

## Acknowledgements

The authors are grateful to Dr. Elliott Kieff (Harvard Medical School, Boston, MA), Dr. Ari Melnick (Weill Cornell Medicine, New York, NY), Dr. Mariusz Wasik

(University of Pennsylvania, Philadelphia, PA), Dr. Martin Carroll (University of Pennsylvania, Philadelphia, PA), Dr. Jianxin You (University of Pennsylvania, Philadelphia, PA) for kindly providing cell lines and reagents. We thank Dr. Zhi Wei, Dr. Tian Tian, and Xiang Lin (New Jersey Institute of Technology, Newark, NJ) for technical support with RNA-Seq analysis, Dr. Huaishan Wang (University of Pennsylvania, Philadelphia, PA) for technical support with mice experiments. We express our thanks to Dr. Stephen Schuster (University of Pennsylvania, Philadelphia, PA), Dr. Robert Yarchoan (National Cancer Institute, Bethesda, MD), Dr. Pat Morin (University of Pennsylvania, Philadelphia, PA), and Dr. Mariusz Wasik (Fox Chase Cancer Center, Philadelphia, PA) for critically reading the manuscript and providing valuable suggestions. This work was supported by the National Cancer Institute at the National Institutes of Health public health service Grants P30-CA016520, R01-CA171979, P01-CA174439, and R01-CA177423 to ESR, the Commonwealth of Pennsylvania through the Baruch S. Blumberg Institute to YD and the Abramson Comprehensive Cancer Center Director fund.

## Author contributions

Y.P., Y.D., and E.S.R. participated in the study design. Y.P., N.H., F.L., L.Z., J.H.W., R.K.S., and H.C.J. performed the experiments. N.H. and Y.D. designed and generated the compounds. Y.P., Y.D., and E.S.R. analyzed these results. Y.P., Y.D., and E.S.R. wrote the manuscript. W.S.E. participated in the interpretation and discussion. All authors reviewed and approved the manuscript.

## Competing interests

The authors declare no competing interests.
