## [Peer Review File · Communications Biology]

Reviewers' comments:

Reviewer #1 (Remarks to the Author):

This interesting manuscript describe Brusatol, a natural quassinoid compound, and several analogs as possible therapies for hematologic malignancies. They report that these compounds, and especially the analogs, are active against a number of tumor lines but have relatively little toxicity. They also do a number of elegant studies showing that a major target for these drugs is PI3K-gamma.

Major comment:

1. They claim that Brusatol and especially the analogs are at least as active but less toxic than the 3 FDA-approved inhibitors (Copanlisib, Duvelisib, and Idelalisib). While it is clear that they have activity, the data about relative toxicity is not as clear. This data is provided particularly in Fig 6a,b and S3 g-i. The way the data is presented, it is difficult in some cases to tell which line is which drug. In Fig 6 a and b, it looks like that analogs (e.g. #15) are if anything more toxic, and no comparisons are shown in mice. Also, in Fig S3g, it is hard to say that the analogs are less toxic, in part because the lines and symbols overlap and in part because the mice start from different weights. Also, no statistics are provided to support the assertion of differential toxicity. This data should be presented more clearly, and if the authors want to claim less toxicity, they should show clearly (with statistics) that this is the case.
2. Since these drugs are active in B cell tumors, it would be of interest to assess their toxicity in normal B cells.

Minor comments:

1. Figure 1a and b. It was not clear why they compared Brusatol and JQ1 at the same doses. Unless they have similar toxicities, it is not clear how meaningful this is.
2. Line 167 and Figure 3a. It is stated that K562 cells were "minimally sensitive" to Brusatol, but this line seems reasonably sensitive with an IC50 of about 0.01 μ M. Perhaps it can be described in a more nuanced way in the text.
3. Figure 3b. It isn't clear what statistics were performed; this should be stated in the Fig legend. It would help to do this throughout.)
4. Line 204. The reader should be directed to the data for reduced toxicity. See the major comment.
5. Line 206. I don't believe compound #1 is described anywhere. This should be done.
6. Figure 4e. What was the statistical comparison between 31 and control?
7. Figure 5c. This data should have a positive control of a sensitive cell line shown previously for comparison.
8. Line 257 and Figure 5i. The description of Fig 5i in line 257 is not clear; what is meant by "the AKT1 expression recovered in the knock-out cell lines with Brusatol of #15 treatment"? Also, in Fig 5i, it looks like compound #15 substantially reduced mTOR and AKT1, but in the text (259) it says these were upregulated. This section should be looked over and clarified.
9. The section from line 309 – 333 in the Discussion seemed to be a detour that did not add too much. It is suggested that the authors review this and if they want to keep it, tie it in better with the rest of the paper. Also, the statement in lines 309-310 should be referenced.
10. There were a number of places that were not clear:
 - a. Line 38: It wasn't clear what was meant by "amelioration of their associated toxicities"
 - b. Fig 1e: What sort of cutaneous lymphoma was it?
 - c. Fig 1f and 3c: It should be stated in the legend what the percentages refer to.
 - d. Fig 2d an section around line 125: Specific carbons (e.g. C3) are referred to, but nowhere are they labeled.

Reviewer #2 (Remarks to the Author):

This is a significant study that makes the important identification of the target of the drug, Brusatol, as PI3Kgamma. Additionally, new derivatives are synthesized and tested and shown to have increased efficacy. The data are convincing and clearly presented. One clarification would be the inclusion of Brusatol in Figure 4 for comparison with the newly derived compounds. Additionally, is it possible to examine the effects on some of the downstream targets in the tumors that develop in the treated mice? If sensitive cell lines are treated with increasing levels of drugs, does resistance develop that maps to PI3K or a downstream target?

Reviewer #3 (Remarks to the Author):

This manuscript presented by Pei et al. described their research on Brusatol and its analogs in treating hematologic malignancies. Brusatol is natural compound that presents anti-tumor activity. While its cellular target thought to be Nrf2 pathway, until now the research field is not entirely convinced that it directly modulates Nrf2 and its MOA remains to be elucidated. Pei et al performed RNAseq and mass spec analyses and identified a protein PI3Kγ as the direct target for Brusatol in lymphoma cell lines. This is an interesting finding since PI3K plays a critical role in signaling network for cell survival and proliferation. The authors further developed several Brusatol analogs and characterized their efficacy and toxicity in cells and in xenograft mouse model. Overall this manuscript is trying to make novel understanding of Brusatol MOA which is interesting to readers. However, it needs significant revision in data quality and explanation. Additional unambiguous evidences are needed to support the authors' conclusion.

Comments:

- 1) The authors make initial connection between Brusatol and PI3K via RNAseq. How long has cells being treated with Brusatol before sequencing? How to confirm changes in PI3K family associated genes are due to Brusatol directly inhibiting PI3K, or merely a secondary effect in inhibiting translation machinery, especially seeing broad down-regulation in PI3K, Akt1, P53 protein levels? Does Brusatol work as a degrader? Similarly, how long has cells being treated with Brusatol before western blotting (Fig 1h-i)?
- 2) Target deconvolution experiment using Brusatol derivative and mass spec provides interesting but not convincing results. How many replicates were performed in this study? Experiment needs another control which uses parent molecule to compete molecule targets from binding to biotinylated molecule.
- 3) Supplemental table 2 has a list of identified proteins, but most of them are only identified by one peptide which is a weak evidence. Experiment needs to be repeated using more starting material/better protocol to increase signal and provides unambiguous protein assignment.
- 4) No peptide sequence was provided in Supplemental table 2 and I'm not clear what score was used here for protein (in excel it refers to "sum of the ion scores of all peptides" – please define ion scores – was it from mascot database search?). Authors should provide raw mass spec data from these studies and write exact parameters used in data acquisition and following data analysis (e.g. database search, peptide identification, FDR filtering etc.). Annotated MS/MS for the PI3K hit is needed in the main figure to provide direct confirmation.
- 5) Does Brusatol and analogs inhibit PI3Kγ in in vitro kinase assay? Perhaps the authors can also test isoform selectivity on various PI3Ks.
- 6) Compound 51048 and 51046 have very equivalent potency in various cell lines, why call one inactive and one active?

- 7) I don't get why the authors call Brusatol treatment minimally affected HL60 and K562 as IC50 for these two cell lines are at 10-20nM range. It also seems like there is disconnection between figure 3a and 3b, since the concentration in 2b is 100nM, which should result in >90% killing for K562.
- 8) Please indicate number of replicates wherever error bars are presented in all figures.
- 9) In figure 1a, 1, 2f, 3b, 4f, please define NS, *, **, *** and ****, and indicate clearly what tests were done between which groups. E.g. In figure 3b it's not clear which two groups were tested statistically.
- 10) Raji cell treated for 24 hours has nearly 1.0 relative viability in Fig 3b. However same concentration treated for 12hrs in Fig 3h only has 0.4 relative viability. Why such a large difference?
- 11) Page 11 refers inactive compound 1, but no structural information was given for this compound. The same page also refers supplemental fig s2b, but this is a wrong figure to be referenced.
- 12) The authors claimed novel Brusatol analogs have enhanced efficacy in cells, but comparing fig 1d and fig 4a, they are quite comparable. Please perform statistical tests to support this claim if any. What more confusing is when comparing fig 4c and 1c, Brusatol clearly is much more potent than these analogs. In xenograft model, is there a statistical difference between brusatol and #15 for their effect on tumor size (fig 4def)? It's not accurate to claim '#15 analog exhibited much stronger inhibition on xenografts than Brusatol' (page 17).
- 13) Why growth inhibition (cytotoxicity) is not dose-dependent for #26 in supp fig 3h? (ctrl = slowest, and then 8mg/kg, 4mg/kg ...)
- 14) In vitro GST-PI3K pull down assay uses 300uM and 500uM which seems to be extremely high comparing to its cellular IC50 of nano molar potency. Why is it?
- 15) Does knock-down PI3K γ (Raji KO cells) make cell Brusatol-resistant?
- 16) I don't agree with the authors on the interpretation of figure 6a/b. Cytotoxicity kills normal cells and clearly Brusatol, #15, #26 all have lower cell viability as compared to clinical PI3K inhibitors. The authors' conclusion that brusatol analogs have less cytotoxicity is incorrect in my opinion.
- 17) Why pan-PI3K inhibitors are less effective than Brusatol if targeting PI3K provides therapeutic effect? Does it suggest PI3K is not the key driving factor and Brusatol & analogs inhibit other critical proteins in these lymphoma cell lines?
- 18) In figure 1A, why an increase in cell number also increases cell viability in such a dramatic way, especially for JQ1?
- 19) Page 20, mass spec section indicates 20ug -> is it actually 20mg? 200 million cells should yield milligram proteins rather than microgram.

Quassinoid analogs with enhanced efficacy for treatment of hematologic malignancies target the PI3K γ isoform (COMMSBIO-19-1344-T)

Dear Editor,

Thank you for your time and efforts serving as our editor. I would also like to thank the reviewers for the valuable time and candid comments towards improving our manuscript. Moreover, we have studied the reviewers' comments closely and agree that the comments were very important and will improve the overall strength of the manuscript. Therefore, we have addressed and incorporated the overall suggestions and specific comments in the revised version. We now think that these changes have substantially improved our manuscript. Finally, we would like to thank the reviewers again for the gracious comments and the editor for handling our manuscript.

Specific responses to the reviewers:

Reviewer #1 (Remarks to the Author):

This interesting manuscript describe Brusatol, a natural quassinoid compound, and several analogs as possible therapies for hematologic malignancies. They report that these compounds, and especially the analogs, are active against a number of tumor lines but have relatively little toxicity. They also do a number of elegant studies showing that a major target for these drugs is PI3K-gamma.

Response: Thanks very much for your encouragement.

Major comment:

1. They claim that Brusatol and especially the analogs are at least as active but less toxic than the 3 FDA-approved inhibitors (Copanlisib, Duvelisib, and Idelalisib). While it is clear that they have activity, the data about relatively toxicity is not as clear. This data is provided particularly in Fig 6a,b and S3 g-i. The way the data is presented, the is difficult in some cases to tell which line is which drug.

In Fig 6 a and b, it looks like that analogs (e.g. #15) are if anything more toxic, and no comparisons are shown in mice. Also, in Fig S3g, it is hard to say that the analogs are less toxic, in part because the lines and symbols overlap and in part because the mice start from different weights. Also, no statistics are provided to support the assertion of differential toxicity. This data should be presented more clearly, and if the authors want to claim less toxicity, they should show clearly (with statistics) that this is the case.

Response: Thanks for your comment. We have revised the statement to clearly show these analogs are more active than the approved inhibitors on several tumor cell lines (Fig 6, d, e, and f). Also, Brusatol and its analogs (#15, #26) did not exhibit any significant cytotoxicity when compared to these approved inhibitors in normal human PBMC, normal T-cells and B-cells (Fig 6, a, b and c). Moreover, our following in vivo toxicity test demonstrated that #26 has the least toxicity among the three tested drugs (Brusatol, #15, and #26), while #15 shows the obvious toxicity in vivo (Fig 6i). We modified the presentation of the indicated Figures (Fig 6, a-f), and replaced the supplementary Figures (Fig S3, g-i) with new figures related to the toxicity test in vivo. The statistical analysis is also included to support our statements.

2. Since these drugs are active in B cell tumors, it would be of interest to assess their toxicity in normal B cells.

Response: Thanks for your comment. As suggestions, we tested these analogs in normal B-cells (Fig 6c). The results showed these compounds had comparable cytotoxicity with these approved drugs in normal B-cells, which is similar to our previous experiments in PBMC and normal T-cells (Fig 6, a and b).

Minor comments:

1. Figure 1a and b. It was not clear why they compared Brusatol and JQ1 at the same doses. Unless they have similar toxicities, it is not clear how meaningful this is.

Response: Thanks for your comment. JQ1 was a potent inhibitor of the bromodomain proteins and a promising drug for the treatment of multiple diseases including hematologic malignancies when we processed this project. Therefore, we used this drug as a positive control to compare the efficacy of our compounds to this new inhibitor JQ1. We should

also note that JQ1 may no longer be used in clinical trials because of its short half-life. We have revised our figures accordingly to highlight the efficiency of our drugs (Fig 1, a, b, and d; Fig 5a) without confusing data from JQ1 as suggested by the reviewer.

2. Line 167 and Figure 3a. It is stated that K562 cells were “minimally sensitive” to Brusatol, but this line seems reasonably sensitive with an IC50 of about 0.01 μ M. Perhaps it can be described in a more nuanced way in the text.

Response: Thanks for your suggestion. We agree with your opinion. The first aim of this experiment was to identify two groups of cell lines, which are sensitive and resistant to Brusatol treatment, respectively. Among more than 20 hematologic malignancies cell lines we tested, the sensitivity to Brusatol treatment between HL60/K562 and MOLM14/SU-DHL-4 shows the significant fold change of approximately 15 times on IC50 values. Therefore, we now describe HL60/K562 as “less sensitive cell lines” and Raji/MOLM14/SU-DHL-4 as “more sensitive cell lines”. This description is more rigorous in line with our results.

3. Figure 3b. It isn't clear what statistics were performed; this should be stated in the Fig legend. It would help to do this throughout.)

Response: We appreciate the reviewer's comment. Multiple t-tests were performed in this experiment. Furthermore, we have now checked all of our figures in the manuscript and have included statistical analysis to strengthen our conclusions. The methods are described in the respective figure legends and the “Materials and methods” section of the manuscript.

4. Line 204. The reader should be directed to the data for reduced toxicity. See the major comment.

Response: Thanks for your comment. Our additional new experimental data showed that analog #26 has remarkably reduced toxicity in vivo as compared to Brusatol and analog #15 (Fig 6i). Therefore, we have modified the statement to adjust our new findings from our in vivo studies.

5. Line 206. I don't believe compound #1 is described anywhere. This should be done.

Response: Thanks for pointing this out. Sorry for the oversight of this control analog. We have now included the chemical structure of #1 analog in supplementary figure S3 (Fig S3d).

6. Figure 4e. What was the statistical comparison between 31 and control?

Response: Thanks for your comment. The activity between analog #31 and control was not significant (NS), and so was not labeled. We have highlighted the efficacy of analogs #15 and #26 in this assay.

7. Figure 5c. This data should have a positive control of a sensitive cell line shown previously for comparison.

Response: Thanks for your comment. We repeated this experiment and set SU-DHL-4 cells as a positive control. The conclusion is consistent with previous observations.

8. Line 257 and Figure 5i. The description of Fig 5i in line 257 is not clear; what is meant by “the AKT1 expression recovered in the knock-out cell lines with Brusatol of #15 treatment”? Also, in Fig 5i, it looks like compound #15 substantially reduced mTOR and AKT1, but in the text (259) it says these were upregulated. This section should be looked over and clarified.

Response: Thanks for your comment. The major goal of this experiment was to validate whether PI3K γ is a key player in Brusatol-regulated PI3K/AKT signaling pathway. Our results showed that signals for the downstream AKT1, GSK3, mTOR, and P53 proteins were rescued in the knock-out Raji cells with Brusatol treatment, further demonstrating that the PI3K γ protein isoform is the target of Brusatol. Regarding analog #15, our previous results demonstrated these analogs can also target the PI3K/AKT signaling pathway (Fig 6, g and h). However, whether they have a similar mechanism to Brusatol is still unknown. Therefore, we revised the description to clarify the main findings.

9. The section from line 309 – 333 in the Discussion seemed to be a detour that did not add too much. It is suggested that the authors review this and if they want to keep it, tie it in better with the rest of the paper. Also, the statement in lines 309-310 should be referenced.

Response: Thanks for your comment. We revised this section to be coherent with other paragraphs and included appropriate references.

10. There were a number of places that were not clear:

a. Line 38: It wasn't clear what was meant by "amelioration of their associated toxicities"

Response: Thanks for your comment. The statement is described in detail in the cited reference (Hanker AB, Kaklamani V, Arteaga CL. 2019. Challenges for the Clinical Development of PI3K Inhibitors: Strategies to Improve Their Impact in Solid Tumors. Cancer Discov 9: 482-91). A major limitation of the development of PI3K inhibitors is the common and dose-dependent toxicity in patients, which depends on their PI3K isozyme specificity. It is still challenging to identify the on-target toxicities of the current PI3K inhibitors.

b. Fig 1e: What sort of cutaneous lymphoma was it?

Response: Thanks for your comment. PDX-129 is a B-cell follicular lymphoma (FL). This has been revised and included in the figure legend, as well as the supplementary table 1.

c. Fig 1f and 3c: It should be stated in the legend what the percentages refer to.

Response: Thanks for your comment. The meaning of the percentages is included in the figure legends as per your suggestions.

d. Fig 2d an section around line 125: Specific carbons (e.g. C3) are referred to, but nowhere are they labeled.

Response: Thanks for your comment. The chemical structure of Brusatol was included in Fig 2d. All of the analogs were modified based on the original structure of Brusatol.

Reviewer #2 (Remarks to the Author):

This is a significant study that makes the important identification of the target of the drug, Brusatol, as PI3K γ . Additionally, new derivatives are synthesized and tested and shown to have increased efficacy. The data are convincing and clearly presented.

Response: Thanks for your encouragement.

One clarification would be the inclusion of Brusatol in Figure 4 for comparison with the newly derived compounds.

Response: Thanks for your comment. Most of these experiments in Figure 4 (now in Figure 5 of the revised manuscript) were performed side by side with the results in Figure 1. To organize the story for better clarity, we separated the results into two different figures since we had to also compare the activity of these analogs to the parent Brusatol based on the relative cell viability. The results demonstrated that these analogs showed less activity than Brusatol in Raji cells (IC50 value between Fig 1c and Fig 5c). However, in general, they exhibited similar activities based on the relative cell viability of multiple hematological diseases associated cell lines (especially in Fig 1d and Fig 5a). Furthermore, studies in vivo also supported our conclusions related to the efficacy of Brusatol and its active analogs (Fig 5, d-e; Fig S3h).

Additionally, is it possible to examine the effects on some of the downstream targets in the tumors that develop in the treated mice?

Response: Thanks for your comment. We examined the expression of several downstream targets from our in vivo studies. These results showed that compound #26 was more effective in inhibiting both PIK3CG and GSK3B expression when compared to the other two compounds (Fig S3j). This suggests that these modified analogs may have different mechanisms for their anti-cancer effects from the parent Brusatol. However, this needs further investigation.

If sensitive cell lines are treated with increasing levels of drugs, does resistance develop that maps to PI3Kg or a downstream target?

Response: Thanks for your comment. We investigated the IC50 of these compounds with PIK3CG knock-out Raji cell lines. However, the inhibitory effects after drug treatment were similar between the knock-out and control cell lines. This suggests that Brusatol may target other cellular factors or that knock-out of PIK3CG is not sufficient to render the

cells resistant to Brusatol treatment and other targets may also be involved as seen from our mass spectra data, which is discussed in our revised manuscript.

Reviewer #3 (Remarks to the Author):

This manuscript presented by Pei et al. described their research on Brusatol and its analogs in treating hematologic malignancies. Brusatol is natural compound that presents anti-tumor activity. While its cellular target thought to be Nrf2 pathway, until now the research field is not entirely convinced that it directly modulates Nrf2 and its MOA remains to be elucidated. Pei et al performed RNAseq and mass spec analyses and identified a protein PI3K γ as the direct target for Brusatol in lymphoma cell lines. This is an interesting finding since PI3K plays a critical role in signaling network for cell survival and proliferation. The authors further developed several Brusatol analogs and characterized their efficacy and toxicity in cells and in xenograft mouse model. Overall this manuscript is trying to make novel understanding of Brusatol MOA which is interesting to readers.

Response: Thanks very much for your encouragement.

However, it needs significant revision in data quality and explanation. Additional unambiguous evidences are needed to support the authors' conclusion.

Response: Thanks for your comment. Based on your suggestions, we have now provided additional evidence and further revised the descriptions in the results section to support our findings. Please review the following specific responses as well as the modified manuscript where we have addressed things throughout.

Comments:

1) The authors make initial connection between Brusatol and PI3K via RNAseq. How long has cells being treated with Brusatol before sequencing? How to confirm changes in PI3K family associated genes are due to Brusatol directly inhibiting PI3K, or merely a secondary effect in inhibiting translation machinery, especially seeing broad down-regulation in PI3K, Akt1, P53 protein levels? Does Brusatol work as a degrader? Similarly, how long has cells being treated with Brusatol before western blotting (Fig 1h-i)?

Response: Thanks for your comment. The LCL1 cells used were treated with Brusatol for 72 hours in our RNA-Seq experiment. To be specific, we did not make the initial connection between Brusatol and PI3Ks via RNA-Seq results until we integrated the results of RNA-Seq and Mass Spectra assay. RNA-Seq analysis only provided evidence as to the potential upstream targets after Brusatol treatment. However, we still did not know the direct targets. We highlighted the PI3Ks in Fig 2c, as well as the list of other targets in Supplementary table S2.

To discuss the results of RNA-Seq and Mass Spectra assay in Fig 2, it is almost impossible to infer that the changes of PI3K associated genes are due to Brusatol directly or indirectly targeting PI3Ks and the downstream genes. However, our following experiments clearly showed that PI3K γ was an important protein that was directly targeted by Brusatol. These included western blot analysis of two distinct groups of cell lines, pull-down in vitro assays, and knock-out assays. Additionally, other data did not support Brusatol as the main inhibitor of the translation machinery. First, most eukaryotic translation inhibitors show nonspecific toxicity and have limited therapeutic value as they broadly block protein synthesis. Our toxicity tests demonstrated that Brusatol and its analogs have limited toxicity in human normal cells (Fig 6, a-c), suggesting that they may have limited toxicity and can be developed as a promising drug. Second, the tested cells were treated with Brusatol for 72 hours in the majority of experiments. The interested PI3K/AKT pathway was significantly inhibited on the protein level, but GAPDH expression as the internal control did not show any changes indicating intact translation machinery (Fig 2, h-i; Fig 3e; Fig 4h; Fig 6, g-h). These data indicate that Brusatol is not directly inducing an inhibitory effect of protein translation.

This idea is novel and interesting as we did not look at the role of Brusatol and its analogs as degraders. We observed obvious inhibitory effects on the PI3K/AKT pathway after treating these cells (Raji and LCL1) for 72 hours (Fig 2, h-i). Brusatol may be a degrader but it would hardly be specific just to these proteins (e.g., PI3K γ , AKT1, P53, GSK3, and Cyclin D1). At this point, we cannot provide clear data and it deserves further investigation.

2) Target deconvolution experiment using Brusatol derivative and mass spec provides

interesting but not convincing results. How many replicates were performed in this study? Experiment needs another control which uses parent molecule to compete molecule targets from binding to biotinylated molecule.

Response: Thanks for your comment. MS results in this study are from one experiment. However, we did several pre-experiments before this mass spectra analysis. This data is a representative set which reflects the actual events in our study. Our efforts were then placed on validating the potential target of Brusatol from the datasets by multiple analyses (bioinformatics, in vitro study, in vivo test). These experiments and results provide a convincing set of data that supports our conclusions. Additionally, we have performed a competitive assay as per your suggestion. The results demonstrated that Brusatol only targeted PI3K γ in vitro, and so can specifically mediate the interaction.

3) Supplemental table 2 has a list of identified proteins, but most of them are only identified by one peptide which is a weak evidence. Experiment needs to be repeated using more starting material/better protocol to increase signal and provides unambiguous protein assignment.

Response: Thanks for your comment. We understand your concerns. The identified peptides do not provide certainty that the critical targets of Brusatol-protein interactions, but identify the potential targets which were validated on our following studies. We initially thought that the mass spectra can provide a relatively clear answer. However, several pre-experiments suggested that the potential binding may be less strong. Discussions with our mass spectra technical support, we focused our efforts on the bioinformatics and biochemistry to narrow down the potential targets. That is the extent of our MS data in the manuscript. Furthermore, other experiments, including RNA-Seq analysis, in vitro pull-down assays, CRISPR/Cas9 knock-out, were used to validate the initial findings. These results clearly showed that Brusatol can directly target PI3K γ protein to inhibit cell viability.

4) No peptide sequence was provided in Supplemental table 2 and I'm not clear what score was used here for protein (in excel it refers to "sum of the ion scores of all peptides" – please define ion scores – was it from mascot database search?). Authors should provide

raw mass spec data from these studies and write exact parameters used in data acquisition and following data analysis (e.g. database search, peptide identification, FDR filtering etc.). Annotated MS/MS for the PI3K hit is needed in the main figure to provide direct confirmation.

Response: Thanks for your comment. We have now included the detailed MS protocol in the revised "Materials and methods", and also submitted the raw data to the MassIVE dataset with the accession number of MSV000085067.

5) Does Brusatol and analogs inhibit PI3K γ in in vitro kinase assay? Perhaps the authors can also test isoform selectivity on various PI3Ks.

Response: Thanks for your suggestion. We don't know this answer at this time but plan to test other PI3K isoforms kinase activity.

6) Compound 51048 and 51046 have very equivalent potency in various cell lines, why call one inactive and one active?

Response: Thanks for your comment. We agree that it is not accurate to describe these biotin-conjugated compounds as inactive or active. We have changed the statement to support our findings. What we initially thought is to perform the MS assay by using active but structurally different compounds, which would help us filter the non-specific interactions and so narrow down the potential targets. Previous studies suggested that the active domain of Brusatol maybe in the proximity of the C-21 position. Therefore, after testing several synthesized compounds, we decided to utilize C-3 modified 51048, 51052 as the positive control, and C-21 modified 51046 as the negative control. This would identify the potential targets that only bind to the C-21 active domain of Brusatol, but no other domains.

7) I don't get why the authors call Brusatol treatment minimally affected HL60 and K562 as IC₅₀ for these two cell lines are at 10-20nM range. It also seems like there is disconnection between figure 3a and 3b, since the concentration in 2b is 100nM, which should result in >90% killing for K562.

Response: Thanks for your careful review. The first aim of this experiment is to find two groups of cell lines, which are sensitive or resistant to Brusatol treatment, respectively. Among more than 20 tested cell lines associated with hematologic malignancies, the sensitivity to Brusatol treatment between HL60/K562 and MOLM14/SU-DHL-4 shows the significant fold change of approximately 15 times by IC50. Therefore, we described these two groups now as “Brusatol-less sensitive” and “Brusatol-more sensitive” cell lines, respectively.

The results in Fig 3a and 3b are from two independent experiments. For K562 cells, treated with 100nM of Brusatol for 72 hours resulted in more than 90% killing (Fig 3a), while another independent experiment showed approximately 50% killing (Fig 3b). The data do not perfectly match in every independent assay because the test system is very sensitive to cell growth and age. Nevertheless, the cell viability is still comparable and supports our conclusions in the same batch of this assay.

8) Please indicate number of replicates wherever error bars are presented in all figures.

Response: Thanks for your comment. We double-checked all of the figures in our manuscript and now included statistical analysis to strengthen our conclusions. These numbers are included in the figures for the revised manuscript. The methods are also described in the respective figure legends as well as the “Materials and methods” section.

9) In figure 1a, 1, 2f, 3b, 4f, please define NS, *, **, *** and ****, and indicate clearly what tests were done between which groups. E.g. In figure 3b it's not clear which two groups were tested statistically.

*Response: Thanks for your comment. We have revised all of the figures accordingly. The labels “NS, *, **, *** and ****” are also now described in the figure legends or the “Materials and methods” section as suggested.*

10) Raji cell treated for 24 hours has nearly 1.0 relative viability in Fig 3b. However same concentration treated for 12hrs in Fig 3h only has 0.4 relative viability. Why such a large difference?

Response: Thanks for your comment. We have removed Fig 3h as we do not have an explanation for such a large difference.

11) Page 11 refers inactive compound 1, but no structural information was given for this compound. The same page also refers supplemental fig s2b, but this is a wrong figure to be referenced.

Response: Thanks for pointing this out. The chemical structure of compound #1 is now included (Fig S3d), and the referred figure is also now referenced in the text.

12) The authors claimed novel Brusatol analogs have enhanced efficacy in cells, but comparing fig 1d and fig 4a, they are quite comparable. Please perform statistical tests to support this claim if any. What more confusing is when comparing fig 4c and 1c, Brusatol clearly is much more potent than these analogs. In xenograft model, is there a statistical difference between brusatol and #15 for their effect on tumor size (fig 4def)? It's not accurate to claim '#15 analog exhibited much stronger inhibition on xenografts than Brusatol' (page 17).

Response: Thanks for your detailed comment. As far as the efficacy, Brusatol and its two analogs (#15, #26) may be quite comparable in the tested hematologic malignancies cells. The recent toxicity test in vivo showed #26 compound had significantly reduced remarkable toxicity in vivo when compared to Brusatol and #15 compound (Fig 6i). Therefore, we modified the statement to highlight the toxicity change without significant efficacy loss.

13) Why growth inhibition (cytotoxicity) is not dose-dependent for #26 in supp fig 3h? (ctrl = slowest, and then 8mg/kg, 4mg/kg ...)

Response: Thanks for your comment. We performed additional experiments to test the toxicity of Brusatol, #15, and #26 in vivo (Fig 6i). The updated results are included in the revised manuscript. The indicated Fig S3h is removed.

14) In vitro GST-PI3K pull down assay uses 300uM and 500uM which seems to be extremely high comparing to its cellular IC50 of nano molar potency. Why is it?

Response: Thanks for your comment. The actual dose is dependent on the specific interactions of compounds and proteins. We firstly performed the binding pull-down assays by incubating 0/20/50/100/300/500 μ M of Biotin-conjugated compounds with the lysates from several cell lines. The experiments indicated that the binding can be detected with more than 100 μ M of Biotin-conjugated compounds, although a weak signal. However, the pull-down assays using 300 μ M and 500 μ M showed strong signals. Similar concentrations were used in other studies (Konze KD, et. al, ACS Chem Biol. 2013).

15) Does knock-down PI3K γ (Raji KO cells) make cell Brusatol-resistant?

Response: Thanks for your comment. We also tested the IC₅₀ of these compounds with PIK3CG knock-out cell lines. The inhibitory effects after drug treatment were similar in knock-out and control cell lines. This suggests that Brusatol may have additional targets including PI3K γ protein and that knock-out of PIK3CG is not sufficient for the cells to be resistant to Brusatol. We also discuss this in the revised manuscript.

16) I don't agree with the authors on the interpretation of figure 6a/b. Cytotoxicity kills normal cells and clearly Brusatol, #15, #26 all have lower cell viability as compared to clinical PI3K inhibitors. The authors' conclusion that brusatol analogs have less cytotoxicity is incorrect in my opinion.

Response: Thanks for your comment. We agree that the cytotoxicity assays are not sufficient to support the conclusion that the analogs are less cytotoxicity as there are no differences between the analogs and clinically approved drugs. To further explore the question, we now performed the toxicity test in vivo with Brusatol and its two analogs (#15 and #26). The results showed that analog #26 was less toxic than the parent Brusatol and compound #15, which suggests that analog #26 may have the potential for further development into clinics (Fig 6i).

17) Why pan-PI3K inhibitors are less effective than Brusatol if targeting PI3K provides therapeutic effect? Does it suggest PI3K is not the key driving factor and Brusatol & analogs inhibit other critical proteins in these lymphoma cell lines?

Response: Thanks for your comment. It should be noted that the tested cell lines selected were sensitive to Brusatol treatment. Although these pan-PI3K inhibitors are less effective than Brusatol in these cell lines, they do exhibit the inhibitory effects with the IC50 values between 1-10 μ M. When compared to the designed pan-PI3K inhibitors, Brusatol a type of natural compound may have other important targets, which is also suggested by our results. Nonetheless, the IC50 values (less than 10 μ M for these tested drugs) indicate a critical role for PI3K isoforms in these cell lines. Besides, the increased efficacy of Brusatol compared to the pan-PI3K inhibitors also demonstrates that Brusatol can target other proteins to inhibit cell viability.

18) In figure 1A, why an increase in cell number also increases cell viability in such a dramatic way, especially for JQ1?

Response: Thanks for your comment. We used the CellTiter-Glo® Luminescent Cell Viability Assay Kit from Promega in this experiment. As described in the manual, it is “a homogeneous method of determining the number of viable cells in culture based on quantitation of the ATP present, an indicator of metabolically active cells”. To further support our findings, we again repeated these assays and had replaced the previous figure with a new one, which also shows similar results.

19) Page 20, mass spec section indicates 20ug -> is it actually 20mg? 200 million cells should yield milligram proteins rather than microgram.

Response: Thanks for your careful review. This has been changed to reflect the correct amount as indicated by the reviewer and also match our records. Thanks for pointing this out.

Thank you for your time, efforts and suggestions to improve our manuscript. I hope that the appropriate changes are now acceptable. We believe that the reviewers' comments have helped us to significantly improve the manuscript.

With sincere regards,

Reviewers' comments:

Reviewer #1 (Remarks to the Author):

This manuscript is substantially improved from the previous submission, and many of the concerns have been addressed. It is a potentially important piece of work. However, I still have two major concerns at this time and several minor concerns:

1. Major concerns

a. The paper is focused on Brusatol and analogs of Brusatol. Three other PI3K inhibitors are approved, but Brusatol is not. For readers who do not know this field intimately, it would help to say something about the problems in the clinical development of Brusatol and whether their new compounds would address these specific problems.

b. In a number of places, the language in the text is inaccurate or does not give all the needed information, and it should be revised. In particular:

i. Line 44 – they should clarify in the text that the study was in xenografts. It sounds like it was in patients, which is confusing since Brusatol did not enter clinical trial

ii. Line 219 – it would be clearer to say that AKT1 was not decreased in the Brusatol-treated cells

iii. Line 240 – They should clarify that all these results were with one dose – 100 μ M

iv. In the general discussion of the analogs (line 364-66), they say that the analogs had significant inhibition of these cancer cells but minimal toxicity. This is not really accurate. The minimal toxicity seems to be mostly from Figs 6a and b, but this is only in PBMC and T cells, and there is a suggestion they may be a bit more toxic. The main evidence for less toxicity seems to be Fig 6i, but here, #26 seems less toxic (although only in 4 mice), but #15 is quite toxic. As stated in the text, the language is somewhat misleading. (Interesting, the analogs appear to be more active than the approved drugs on a molar basis in 6d-f).

2. A couple of minor points

a. On line 85, Figure 1F is mislabeled as 1d

b. In Figure 6i, the Kaplan-Meier lines are overlapping and it is hard to see what is happening. They should somehow be separated.

Reviewer #2 (Remarks to the Author):

The authors have carefully responded to the concerns of the previous reviewers. The revised manuscript presents the data more clearly and includes additional data requested by the reviewers. Importantly, the MS data and the derived conclusions are verified by additional bioinformatic and experimental approaches. The requested additional experimental details including replicates and MS data are not included. The IPA data are particularly clearly presented. It will be interesting what other potential molecules or pathways are also targeted by brusatol.

Reviewer #3 (Remarks to the Author):

Comments

The revised manuscript is much improved and has mostly cleared my concerns. Outstanding questions:

1) In figure 4b, band intensity without a control to compare against is not meaningful. Authors should include SU-DHL-4 on the same blots with three NPC cell lines to help readers interpret the PI3K γ difference.

2) In figure 4h, every band in the sgVec + Brusatol lane is disappeared, including GAPDH loading control. In fact, most bands in sgVec-Brusatol lane look almost identical, the authors should carefully check the authenticity of this data. This might be an accidental editing error. One of the supplemental files with full size blot images for figure 4h has a different view.

Quassinoid analogs with enhanced efficacy for treatment of hematologic malignancies target the PI3K γ isoform (COMMSBIO-19-1344A)

Dear Editor,

Thank you very much for your time and efforts. I would also like to thank the reviewers for the valuable time and candid comments towards improving our manuscript. We agree that these comments or suggestions are very important to improve our manuscript. Therefore, we have addressed and incorporated the specific comments in the revised version. Finally, we thank the reviewers again for the gracious comments and the editor for handling our manuscript.

Specific responses to the reviewers:

Reviewer #1 (Remarks to the Author):

This manuscript is substantially improved from the previous submission, and many of the concerns have been addressed. It is a potentially important piece of work.

Response: Thanks for your encouragement.

However, I still have two major concerns at this time and several minor concerns:

1. Major concerns

a. The paper is focused on Brusatol and analogs of Brusatol. Three other PI3K inhibitors are approved, but Brusatol is not. For readers who do not know this field intimately, it would help to say something about the problems in the clinical development of Brusatol and whether their new compounds would address these specific problems.

Response: Thanks for your comments. We briefly introduce the main obstacle for the clinical applications of Brusatol in the second paragraph of “Introduction”. This refers to its unclear anti-cancer mechanisms and its associated toxicities, which are the two targets of our study.

b. In a number of places, the language in the text is inaccurate or does not give all the needed information, and it should be revised.

Response: Thanks for your comments. We double-checked the manuscript and further improved its accuracy, which was included in the revised version with track changes.

In particular:

i. Line 44 – they should clarify in the text that the study was in xenografts. It sounds like it was in patients, which is confusing since Brusatol did not enter clinical trial

Response: Thanks for your comments. We describe the statement as “One study showed that Brusatol can be used as an adjuvant chemotherapeutic drug in A549 lung cancer cells derived xenografts by inhibiting the Nrf2 signaling pathway.”

ii. Line 219 – it would be clearer to say that AKT1 was not decreased in the Brusatol-treated cells

Response: Thanks for your comments. We agree with your opinion and have revised it accordingly.

iii. Line 240 – They should clarify that all these results were with one dose – 100 μ M

Response: Thanks for your comments. We highlighted the procedure as “... Brusatol or associated analogs were added once to the cells without media change” in “Cell viability assay” of “Materials and methods”, which could summarize all of the procedures in our study.

iv. In the general discussion of the analogs (line 364-66), they say that the analogs had significant inhibition of these cancer cells but minimal toxicity. This is not really accurate. The minimal toxicity seems to be mostly from Figs 6a and b, but this is only in PBMC and T cells, and there is a suggestion they may be a bit more toxic. The main evidence for less toxicity seems to be Fig 6i, but here, #26 seems less toxic (although only in 4 mice), but #15 is quite toxic. As stated in the text, the language is somewhat misleading. (Interesting, the analogs appear to be more active than the approved drugs on a molar basis in 6d-f).

Response: We appreciate the reviewer's comments. The statements have been revised to demonstrate our results as your suggestions.

2. A couple of minor points

a. On line 85, Figure 1F is mislabeled as 1d

Response: Thanks for your comments. It has been corrected in the revised manuscript.

b. In Figure 6i, the Kaplan-Meier lines are overlapping and it is hard to see what is happening. They should somehow be separated.

Response: Thanks for your suggestions. The revised figure has been included in the manuscript.

Reviewer #2 (Remarks to the Author):

The authors have carefully responded to the concerns of the previous reviewers. The revised manuscript presents the data more clearly and includes additional data requested by the reviewers. Importantly, the MS data and the derived conclusions are verified by additional bioinformatic and experimental approaches. The requested additional experimental details including replicates and MS data are not included. The IPA data are particularly clearly presented. It will be interesting what other potential molecules or pathways are also targeted by brusatol.

Response: Thanks very much for your encouragement. In this study, we explored the anti-cancer activities of Brusatol and developed its analogs as the potential therapeutic agents for treatment of hematologic malignancies. We hope this can further contribute to the development of PI3K inhibitors, and we are continuing to fully reveal the anti-cancer mechanisms of this classical natural drug.

Reviewer #3 (Remarks to the Author):

The revised manuscript is much improved and has mostly cleared my concerns.

Response: Thanks very much for your encouragement.

Outstanding questions:

1) In figure 4b, band intensity without a control to compare against is not meaningful. Authors should include SU-DHL-4 on the same blots with three NPC cell lines to help readers interpret the PI3K γ difference.

Response: Thanks for your suggestions. After reviewing our primary data, we found that the blots of SU-DHL-4 cells were determined in the same experiment, which was not shown in this figure because they were not all NPC cells. We now included these blots in this revised figure (Fig 4b). And the source data can be reviewed in our submitted supplementary named “Data file S2. The source data in this manuscript (Excel file)”.

2) In figure 4h, every band in the sgVec + Brusatol lane is disappeared, including GAPDH loading control. In fact, most bands in sgVec-Brusatol lane look almost identical, the authors should carefully check the authenticity of this data. This might be an accidental editing error. One of the supplemental files with full size blot images for figure 4h has a different view.

Response: Thanks so much for your careful review and for catching this mistake. This is indeed an editing error and has been corrected in the revised figure. The source data is also included in our supplemental file named “Data file S2. The source data in this manuscript (Excel file)” for your reference. We have also checked other figures to confirm the accuracy of our data.

Thank you again for your time, efforts and suggestions to improve our manuscript. I hope that the appropriate changes are now acceptable. We believe that the reviewers' valuable comments have helped us to significantly improve the manuscript.